# Exploring YAP1-centered networks linking dysfunctional CFTR to epithelial–mesenchymal transition

Margarida C Quaresma, Hugo M Botelho ⓘ, Ines Pankonien ⓘ, Cláudia S Rodrigues, Madalena C Pinto ⓘ, Pau R Costa ⓘ, Aires Duarte, Margarida D Amaral ⓘ

**Mutations in the CFTR anion channel cause cystic fibrosis (CF) and have also been related to higher cancer incidence. Previously we proposed that this is linked to an emerging role of functional CFTR in protecting against epithelial–mesenchymal transition (EMT). However, the pathways bridging dysfunctional CFTR to EMT remain elusive. Here, we applied systems biology to address this question. Our data show that YAP1 is aberrantly active in the presence of mutant CFTR, interacting with F508del, but not with wt-CFTR, and that YAP1 knockdown rescues F508del-CFTR processing and function. Subsequent analysis of YAP1 interactors and roles in cells expressing either wt- or F508del-CFTR reveal that YAP1 is an important mediator of the fibrotic/EMT processes in CF. Alongside, five main pathways emerge here as key in linking mutant CFTR to EMT, namely, (1) the Hippo pathway; (2) the Wnt pathway; (3) the TGFβ pathway; (4) the p53 pathway; and (5) MYC signaling. Several potential hub proteins which mediate the crosstalk among these pathways were also identified, appearing as potential therapeutic targets for both CF and cancer.**

## Introduction

The cystic fibrosis transmembrane conductance regulator (CFTR) protein is a chloride ($Cl^-$) and bicarbonate ($HCO_3^-$) channel expressed at the apical plasma membrane (PM) of epithelial cells in a wide variety of tissues (1). Mutations in the CFTR gene cause cystic fibrosis (CF), a multi-organ disease that can manifest itself through pancreatic insufficiency, intestinal obstruction, liver disease, male infertility and, most importantly, causing respiratory impairment/failure, the leading cause of CF-related morbidity and mortality (2). CF is the most common life-shortening monogenic condition in Caucasians, affecting more than 90,000 individuals worldwide (3). Approximately 80% of these individuals bear the deletion of residue phenylalanine 508 (F508del) on at least one allele (4), making it the most common CFTR mutation.

Besides its classical manifestations, CF has been recently correlated with increased risk of cancer. A comprehensive study of a large cohort of individuals with CF found higher risk of several cancer forms, particularly digestive tract cancer, lymphoid leukemia and testicular cancer (5). In fact, even CF carriers (with ~50% CFTR channel activity versus control individuals) have been described to have a significantly higher risk of gastrointestinal and pancreatic cancer (6). Moreover, several cancer studies have identified CFTR as a tumor suppressor, whereby down-regulated CFTR favors cancer development both in vitro and in vivo. Such associations have been found in several cancer tissues and cells, such as breast (7), lung (8, 9), colon (10, 11), pancreas (12), and prostate (13). CFTR down-regulation has also been associated with poor cancer prognoses (8, 9, 10), malignancy and metastasis (8, 9).

A probable explanation for the role of CFTR as a tumor suppressor is its emerging association to epithelial–mesenchymal transition (EMT) (14). EMT is a cellular process whereby polarized fully differentiated epithelial cells transition into a mesenchymal phenotype. It is an essential developmental process which can be reactivated in certain physiological contexts (namely tissue repair) and pathological conditions (e.g., fibrosis and cancer progression) (15). During EMT epithelial cells lose, to variable degrees, their specialized features, including strong cell–cell junctions, apical-basal polarity, cell shape, and cytoskeleton organization. On the other hand, they gain mesenchymal characteristics like front-rear polarity, cell individualization, motility, and invasive behavior (15, 16). Depending on the biological context, epithelial cells undergoing EMT can co-express epithelial and mesenchymal phenotypes, that is, partial EMT (16, 17). The relationship between CFTR and EMT (reviewed elsewhere (18)) has been suggested based on a direct link between the down-regulation of CFTR in CF or tumor cells and EMT induction. Through transcriptome profiling meta-analysis we identified an EMT signature in the airways of individuals with CF (19) and more recently showed that expression of mutant CFTR is enough to trigger a TWIST1-mediated partial EMT in the CF airways (14). However, the pathways that link CFTR to EMT are still poorly understood.

Thus, our aim here is to elucidate how CFTR, an apical anion channel, can impact such a broad process that can completely change the differentiation state of epithelial cells. We adopted a systems biology approach to identify potential EMT pathways and

BioISI–Biosystems and Integrative Sciences Institute, Faculty of Sciences, University of Lisboa, Lisboa, Portugal

Correspondence: mdamaral@fc.ul.pt

protein networks that might be active/dysregulated in response to dysfunctional CFTR and thus affect the overall cellular phenotype including epithelial differentiation and consequently the secretory trafficking machinery. Combining functional genomics and proteomics, we were able to find that the Hippo-associated protein Yes-associated protein 1 (YAP1) and its transcriptional partner TEA domain transcription factor 4 (TEAD4) are potential key mediators of EMT and fibrosis in CF. We found that YAP1, while interacting with F508del, but not with wt-CFTR, is aberrantly active in the presence of F508del-CFTR and that its knockdown (KD) rescues both F508del-CFTR processing and function, being additive to CFTR corrector drugs. Finally, we identified five key EMT pathways dysregulated in CF cells and several proteins among the F508del-CFTR–specific YAP1 interactors, which emerge as major hubs possibly mediating the crosstalk among these pathways and appearing as potential therapeutic targets for both CF and cancer.

## Results

In this work, we followed a systems biology approach to identify the pathways linking EMT and dysfunctional CFTR (Fig 1). First, we applied functional genomics, namely a siRNA high-throughput screening platform (20) to assess which EMT-associated genes could be down-regulated (KD) to rescue the PM traffic of F508del-CFTR, while not affecting wt-CFTR traffic. This was followed by validation and characterization steps and finally mass spectrometry (MS)-based proteomics to generate protein–protein interaction (PPI) networks highlighting potential EMT/fibrosis/cancer-related pathways in CF (Fig 1). Importantly, this study design (complementing siRNA screens with proteomic data) has already proven a powerful approach in identifying previously unrelated genes/pathways, including components of major EMT-associated pathways such as Wnt/β-catenin (21).

### EMT-related hippo signaling is active in CF cell models

We hypothesized that EMT-related genes whose genetic inactivation had no effect on wt-CFTR traffic but enhanced/rescued F508del-CFTR traffic would likely integrate EMT pathways aberrantly active in response to dysfunctional CFTR. To identify such genes, we applied a previously described high-throughput fluorescence microscopy screening pipeline to quantify CFTR traffic (20) in a hypothesis-driven siRNA screen to target genes known to be associated with EMT and (de-)differentiation. A total of 83 unique siRNAs or siRNA combinations were tested on cystic fibrosis bronchial epithelial (CFBE) mCherry-Flag-F508del-CFTR and wt-CFTR reporter cell lines for their ability to modulate the amount of PM-located CFTR (Supplemental Data 1). Each gene was targeted by at least two different siRNA molecules. A total of 63 traffic hits was obtained, comprising siRNAs which enhance (Z > 1) or inhibit (Z < -1) wt- or F508del-CFTR traffic (Fig 2A). A subset of 24 siRNAs were considered as high confidence hits either because the traffic phenotype was the same for at least two siRNAs or two siRNA combinations targeting the same gene(s) or the traffic hit phenotype was restricted to one siRNA but significant complementary

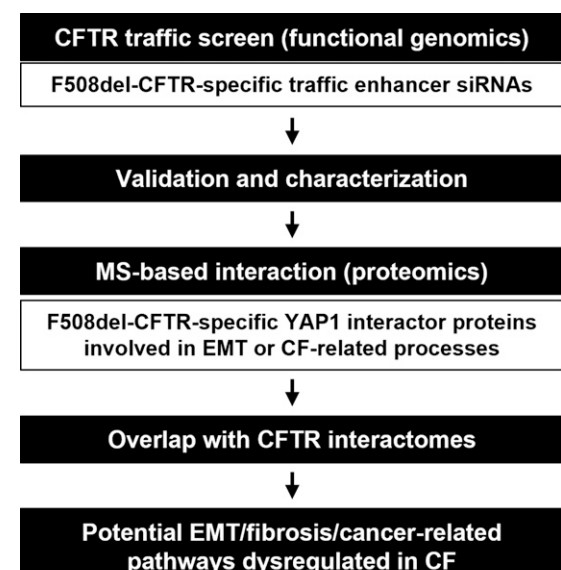

**Figure 1. Schematic representation of the work.**
A systems biology approach was followed to identify the pathways linking epithelial–mesenchymal transition and dysfunctional CFTR. An initial functional genomics approach (an siRNA CFTR traffic screen) was followed by MS-based proteomics to generate protein–protein interaction (PPI) networks highlighting potential epithelial–mesenchymal transition/fibrosis/cancer-related pathways in CF.

data on the target protein was obtained in this work (i.e., differential expression levels in wt- and F508del-CFTR expressing cells, see Supplemental Data 1). A final subset of four top hits (EMT-related gene products whose KD selectively rescues F508del-CFTR traffic) was identified and comprised YAP1, TEAD4, TWIST1, and CEBPB.

The effect of the two siRNAs against YAP1 was the most striking, with both significantly enhancing F508del-CFTR PM traffic (Z = 1.05 and 1.37, Fig 2B and C and Supplemental Data 1). Two different siRNA combinations, namely siYAP1+siTEAD4 also significantly enhanced F508del-CFTR traffic (Z = 1.70 and 1.26, Fig 2B and C and Supplemental Data 1). Both genes are key components of the mammalian EMT/cancer-related Hippo pathway, being YAP1 a major transcriptional regulator (22) and TEAD4 its essential binding partner in gene transcription regulation (23). Importantly, down-regulation of these Hippo pathway elements did not affect wt-CFTR traffic (Fig 2B and C and Supplemental Data 1). In fact, two different combinations targeting siYAP1+siTEAD4 and siTEAD4+siTAZ (another major Hippo player), emerged as high confidence wt-CFTR traffic inhibitors (Supplemental Data 1). These results are suggestive that Hippo signaling (particularly YAP1 and TEAD4) may be specifically dysregulated in F508del-CFTR expressing cells.

Other top hits of the screen were siTWIST1 (1) (Z = 1.37) and siCEBPB (2) (Z = 1.1), both also enhancing the F508del-CFTR PM traffic but not that of wt-CFTR (Fig 2B and D and Supplemental Data 1), albeit the effect being limited to one siRNA. However, given that C/EBPβ, TWIST1, TEAD4, and YAP1 all showed differential protein expression in polarized wt- versus F508del-CFTR expressing cells (Fig 2E and F) we considered these data significant enough to suggest an implication of these genes in EMT in the context of CF. In particular, YAP1 expression levels are significantly increased in

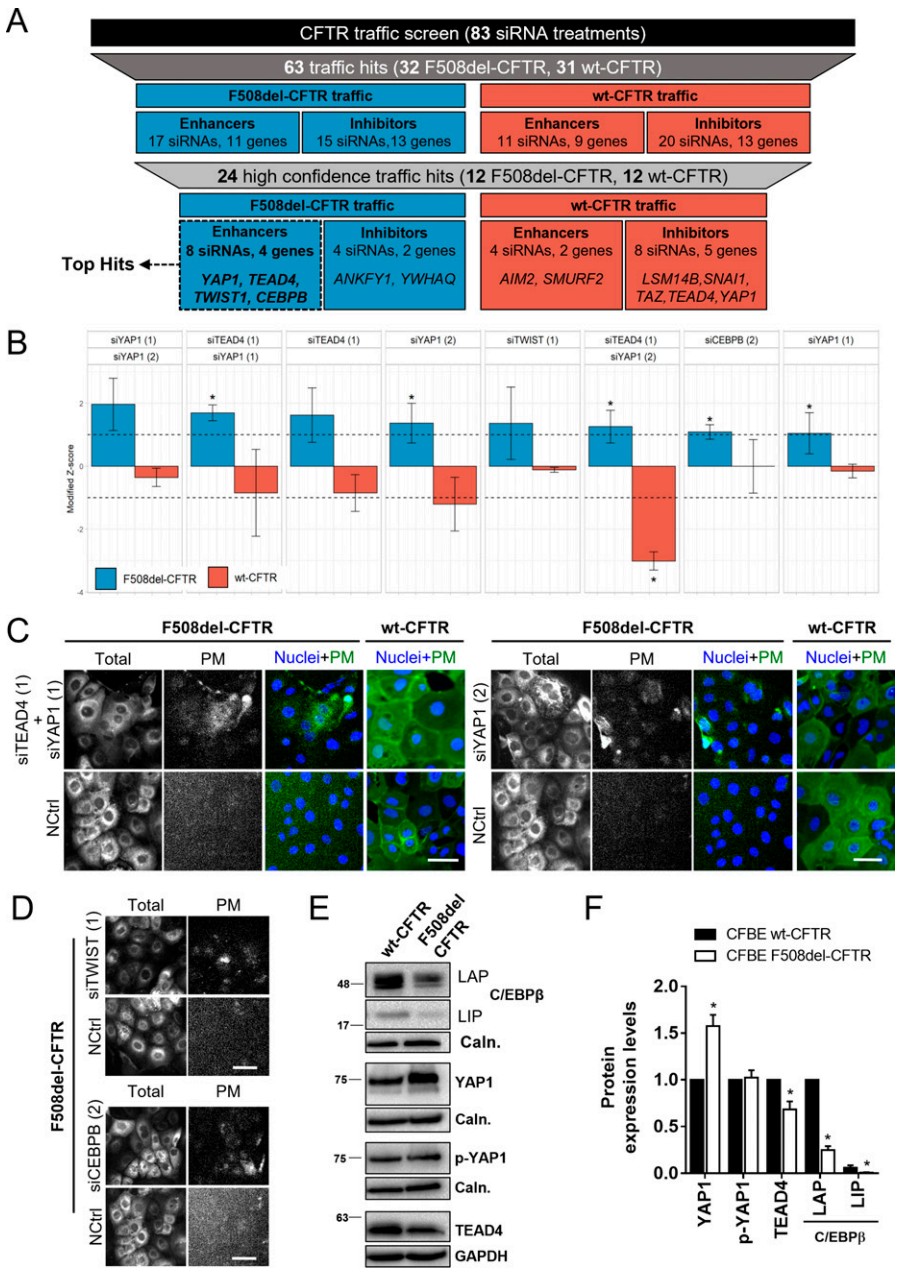

**Figure 2. A CFTR traffic screen identifies YAP1, TEAD4, CEBPB, and TWIST1 as potential factors bridging epithelial–mesenchymal transition and CFTR traffic.**
**(A)** Summary of the siRNA traffic screen results (Supplemental Data 1). A total of 83 different siRNA treatments were tested on Cystic Fibrosis Bronchial Epithelial mCherry-Flag-wt- and F508del–CFTR cells. When assessing plasma membrane (PM) localization 63 traffic hits were obtained, comprising siRNAs which significantly enhance (Z > 1) or inhibit (Z < -1) CFTR traffic versus the baseline. 24 siRNAs were considered high confidence hits and a top hit subset was selected (eight siRNAs targeting four genes) which specifically enhance F508del-CFTR traffic. **(B)** Top hits from the traffic screen and their respective modified Z-score. Immunofluorescence images were quantified to determine the CFTR PM levels in response to each siRNA treatment, which were then converted into a modified Z-Score. Data are presented as median ± SEM. * indicates significant differences between the siRNA treatment and the NCtrl treatment (unpaired *t* test, *P* < 0.05). Numbers within brackets refer to specific siRNA sequences (Supplemental Data 1) (n = 3–4). **(C)** Representative fluorescence microscopy images from top hits of the traffic screen. The mCherry fluorescence represents the total amount of CFTR and the Flag fluorescence represents the amount of CFTR at the PM. Traffic of mCherry-Flag-F508del-CFTR was enhanced by siTEAD4 (1) + siYAP1 (1) and siYAP1 (2) as represented by an increase in Flag (PM) fluorescence. These treatments did not increase mCherry-Flag-wt-CFTR traffic. In merged images nuclei are depicted in blue and Flag (PM CFTR) in green. Scale bar represents 50 *µ*m. **(D)** Representative fluorescence microscopy images from top hits of the traffic screen with effect restricted to one siRNA. Traffic of mCherry-Flag-F508del-CFTR was enhanced by siTWIST1 (1) and siCEBPB (2) as represented by an increase in Flag fluorescence. Scale bar represents 50 *µ*m. **(E)** Representative WB of C/EBP*β*, YAP1, p-YAP1, and TEAD4 protein levels in polarized wt- and F508del-CFTR cystic fibrosis bronchial epithelial cells. **(E, F)** Densitometric quantification of the protein expression detected by WB in (E). Data are normalized to loading control (calnexin, GAPDH) and the protein expression in wt-CFTR cells and showed as mean ± SEM. * indicates significant difference between wt- and F508del-CFTR cells (unpaired *t* test, *P* < 0.05) (n = 3).

F508del-CFTR cells (Figs 2E and F and S1) – although its inactive form levels (phosphorylated YAP1, p-YAP1) are unchanged. In contrast, C/EBP*β* and TEAD4 expression levels are significantly increased in wt-CFTR expressing cells (Fig 2E and F). We have also previously found TWIST1 to be significantly increased in F508del-CFTR expressing cells (14).

To validate the traffic screen findings, we performed both semi-quantitative PCR and Western blot (WB) analyses for the top four hits on a different cell model: CFBE cells expressing wt- or F508del-CFTR. These results show that both siRNAs targeting YAP1, TEAD4 and TWIST1 resulted in a 50–70% reduction of their respective transcript levels (Fig S2A–D) and ~90% KD of protein levels (Figs 3A and B and S2E and F). On the other hand, siRNAs targeting CEBPB

showed no effect on the transcript levels (Fig S2A–D) with screen hit siCEBPB (2) even slightly increasing C/EBP*β* protein levels (Figs 3A and B and S2E and F), suggesting a possible off-target effect in this system. Responses were similar in CFBE cells expressing both F508del- and wt-CFTR. The effect of each siRNA on CFTR processing (i.e., the fraction of the PM-characteristic band C versus total CFTR amount) was evaluated by WB. All top siRNA hits from the traffic screen (siYAP1 [1] and [2], siTEAD4 [1], siTWIST1 [1], and siCEBPB [2]) were also found to modestly but significantly increase F508del-CFTR processing versus the control (Fig 3A and C). In accordance with the screen data, siTEAD4 (2), siTWIST1 (2), and siCEBPB (1) did not significantly affect F508del-CFTR processing, nor did any of the siRNAs targeting the four top hits affect wt-CFTR processing (Fig S2E

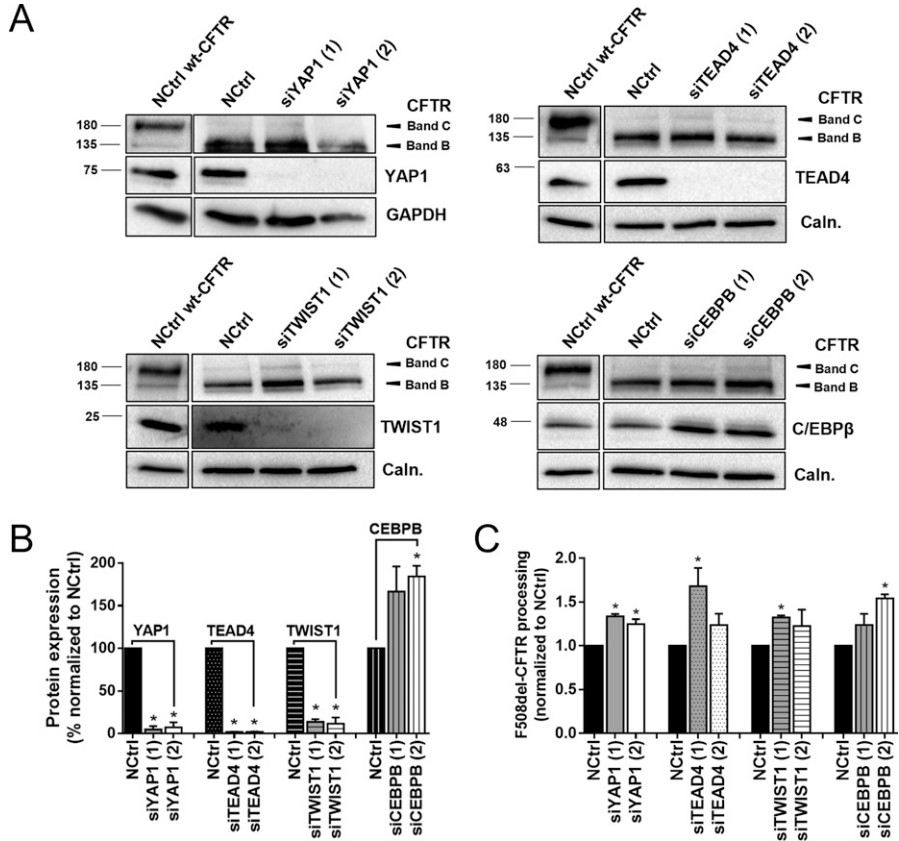

**Figure 3. Hit validation confirms target-specific KD and F508del-CFTR rescue to the plasma membrane.**
**(A)** Representative WB of CFTR, YAP1, TEAD4, TWIST1, and C/EBPβ protein levels in F508del-CFTR cystic fibrosis bronchial epithelial cells treated with a non-targeting siRNA (NCtrl), siYAP1, siTEAD4, siTWIST1, and siCEBPB (1) and (2). The WB blot shows rescue of F508del-CFTR by the appearance of its fully glycosylated form (band C). Protein levels in wt-CFTR treated with NCtrl are also shown for comparison. **(A, B)** Densitometric quantification of YAP1, TEAD4, TWIST1, and C/EBPβ protein expression in (A). Data are normalized to loading control (calnexin and GAPDH) and showed as a percentage (%) of the protein expression of the NCtrl, mean ± SEM. * indicates significant differences between the siRNA treatment and the NCtrl treatment (unpaired $t$ test, $P < 0.05$) (n = 3). **(A, C)** Densitometric quantification of F508del-CFTR processing in (A). CFTR processing is the densitometric band ratio of ratio of fully glycosylated versus total CFTR (C/[B+C]). Data are normalized to the loading control (calnexin and GAPDH) and NCtrl and showed as mean ± SEM. * indicates significant differences between siRNA and NCtrl treatment (unpaired $t$ test, $P < 0.05$) (n = 3).

and G), further suggesting the absence of EMT when functional CFTR is present.

### The EMT transcription factor YAP1 is aberrantly active in response to F508del-CFTR

Among the four top hits of the traffic screen, only protein expression levels of transcription factors (TFs) YAP1 and TWIST1 were significantly increased in F508del- versus wt-CFTR expressing cells, suggesting that any aberrant EMT triggered by F508del-CFTR might be mediated by one or both of these TFs. We thus performed a co-immunoprecipitation experiment to identify whether CFTR interacts with YAP1 or TWIST1. Whereas TWIST1 was found to interact with both wt- and F508del-CFTR (Fig 4A, arrows), YAP1 only interacts with F508del-CFTR (Fig 4A, arrowheads). Importantly, no interaction was detected between YAP1 and TWIST1 (Fig 4A), suggesting an independent transcriptional response activation. Considering the selectivity of the YAP1–F508del–CFTR interaction, as well as the consistent rescue of the F508del-CFTR traffic phenotype in response to YAP1 KD–siYAP1 (1)/(2), siTEAD4 (1) + siYAP1 (1)/(2) – YAP1 was classified as the most promising hit, and we sought to further characterize its role in EMT induction in CF models. To further confirm this, we assessed YAP1 expression in different airway cell models (16HBE and 16HBEge cells) expressing wt-, F508del-, and N1303K-CFTR (an additional CF-causing mutation) and observed that also on these cells, with the presence of dysfunctional CFTR, YAP1 expression levels are significantly increased, regardless of the CFTR mutation (Fig S3A and B).

We thus performed patch-clamp experiments and observed that YAP1 KD also significantly increased F508del-CFTR cAMP–activated current density comparing with controls (Fig 4B and C). Thus, YAP1 KD rescues F508del-CFTR traffic, processing, and function. YAP1 KD also had an additive effect to the treatment with CFTR traffic corrector drugs currently in the clinic, namely, VX-661, VX-809, and VX-445+VX-661 (Fig 4D and E), significantly increasing levels of F508del-CFTR processed form (band C) by 2.5-, 2.9-, and 1.8-fold, respectively. YAP1 KD combined with VX-445+VX-661 treatment in 16HBEge F508del-CFTR cells also significantly increased the levels of band C approximately by 1.8-fold (FigS3C and D), thus confirming the previous observed effect in a different cell model. CFTR modulator drugs did not have a significant effect on YAP1 levels (Fig S3E and F).

Last, we had previously observed that YAP1 expression levels were significantly increased in F508del-CFTR cells, whereas the levels of its inactive form were unchanged between F508del- and wt-CFTR cells (Fig 2E and F), which suggested higher levels of active YAP1 in the presence of dysfunctional CFTR. To further confirm this, we assessed and quantified the subcellular localization of YAP1 in polarized CFBE wt- and F508del-CFTR cells because the nuclear fraction of YAP1 corresponds to its active form. Indeed, it was possible to observe that the nuclear fraction of YAP1 (i.e., the ratio between nuclear and whole cell fluorescence) is significantly increased in F508del- versus wt-CFTR cells (Fig 4F and G). This further confirms higher nuclear accumulation in F508del-CFTR expressing cells, suggesting an abnormal increase in active YAP1 in CF cells.

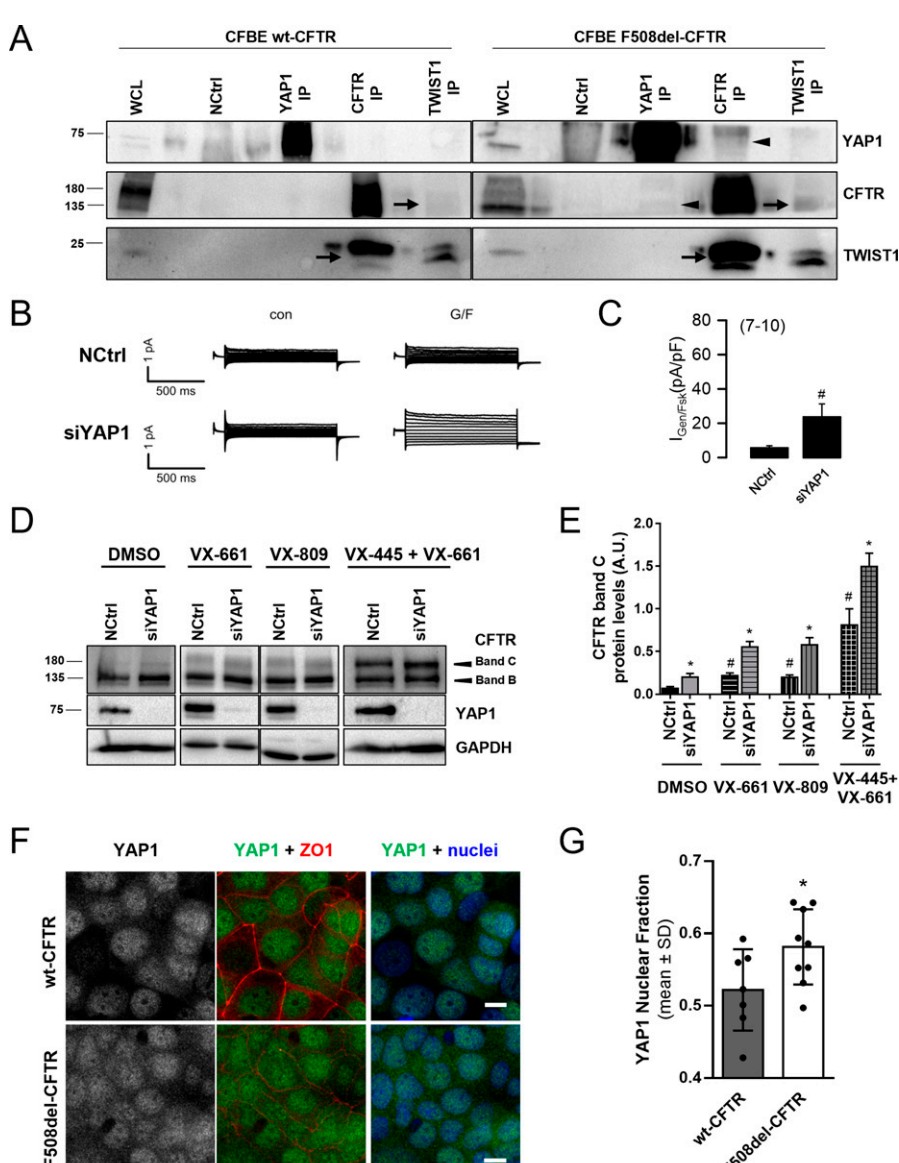

**Figure 4. YAP1 characterization points to its role as an important link between CFTR and epithelial–mesenchymal transition/differentiation.**
**(A)** YAP1, CFTR, and TWIST1 co-immunoprecipitation performed in cystic fibrosis bronchial epithelial (CFBE) wt- and F508del-CFTR cells. A whole cell lysate (WCL) and beads without any antibody (NCtrl) were run in parallel with the pull downs. TWIST1 interacts with both wt- and F508del-CFTR (arrows) whereas YAP1 interacts with F508del- but not wt-CFTR (arrowheads) (n = 3). **(B, C)** Original tracings and (C) quantification of whole-cell patch-clamp measurements of CFTR activity showing functional rescue of F508del-CFTR in CFBE cells after YAP1 down-regulation. Data are shown as mean ± SEM. # indicates significant difference between siYAP1 and non-targeting siRNA (NCtrl) treatments (unpaired t test, P < 0.05) (n = 7–10 cells). **(D)** Representative WB showing the effect of the combination of YAP1 knockdown with the CFTR modulator drugs VX-809, VX-661, and VX-445+VX-661 on rescuing F508del-CFTR processing (on CFBE F508del-CFTR cells). **(E)** Densitometric quantification of the band C protein expression in (D). Data are normalized to the loading control (GAPDH). Data shown as mean ± SEM. # indicates significant differences between NCtrl + DMSO and NCtrl + modulator treatment and * indicates significant differences between siYAP1 and NCtrl treated with the same modulator (unpaired t test, P < 0.05) (n = 3). **(F)** Representative image of the subcellular localization of YAP1 in polarized CFBE cells by immunofluorescence. Images are shown as average intensity projections. In merged pictures, nuclei are depicted in blue, YAP1 in green, and ZO1 in red. Scale bar represents 10 μm (n = 3). **(G)** Quantification of the YAP1 nuclear fraction from the immunofluorescence images, corresponding to the fraction of fluorescence localizing to the nucleus of each cell comparing with whole-cell fluorescence. Each image (represented by individual dots) was summarized by the average nuclear fraction value of all its cells, and global data are showed as mean ± SD. * indicates significant differences between wt- and F508del-CFTR–expressing cells (n = 7–8).

## F508del-CFTR leads to YAP1 interactome remodeling

To characterize the wider protein network linking YAP1 to F508del-CFTR (versus wt-CFTR), YAP1 immunoprecipitation was performed using isogenic CFBE cell lines expressing wt- and F508del-CFTR followed by MS-based interaction proteomics (in three independent replicates). This allowed us to assess how the YAP1 interactome is modified in the presence of F508del-CFTR (versus wt-CFTR), without interference of any other factors (e.g., inflammation and modifier genes). YAP1 pulldown yielded a distinct SDS–PAGE band pattern between the F508del-CFTR and wt-CFTR cell lines (Fig S4A) and, unsurprisingly, several differential YAP1 interactors in F508del- versus wt-CFTR expressing cells were identified by MS (Supplemental Data 2). We considered as the YAP1 interactome the set of proteins identified in at least two biological replicates. It included 214 common interactors (i.e., present in both cell lines),

138 interactors specific to F508del-CFTR cells, and 20 interactors specific to wt-CFTR cells (Fig 5A). Thus, CFTR dysfunction largely increased the number of YAP1 interactors. We then crossed these data with published YAP1 interactomes (24, 25, 26) and identified 79 common proteins (33.8% of our list) in the YAP1 interactors in wt-CFTR cells and 98 common proteins (27.8% of our list) in the YAP1 interactors in F508del-CFTR cells (Fig S4B). These coverages are a good indicator that we have successfully identified specific YAP1 interactomes, but also that we identified novel YAP1 interactors.

To describe the biological pathways occurring in our datasets, we performed a Gene Ontology (GO) analysis to identify enriched biological process (BP) terms (Supplemental Data 3), which we then grouped into manually curated biological categories for more straightforward data visualization (see the Materials and Methods section). We established both common (Fig S5A) and differentially (Fig 5B) enriched BP terms for the YAP1 interactors in wt- and

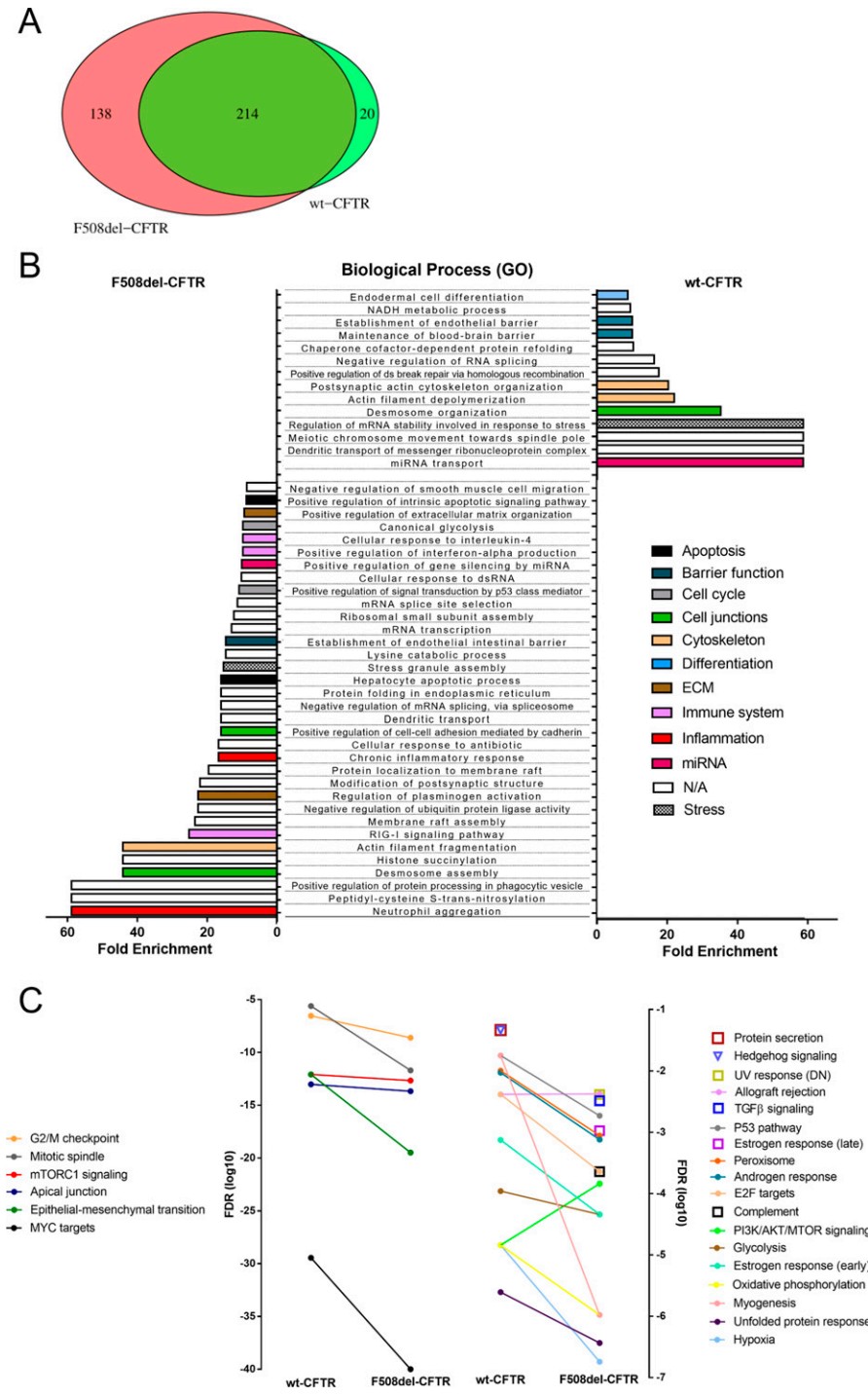

**Figure 5. Gene Ontology analysis and gene set enrichment analysis of YAP1 interactors suggest a specific YAP1-centered epithelial–mesenchymal transition/oncogenesis-mitosis-hypoxia-inflammation signature in the presence of F508del-CFTR.**

**(A)** Venn diagram showing the protein overlap of the YAP1 interactors found in at least two different replicates of wt- and F508del-CFTR cells. 214 common interactors were found, as well as 138 interactors specific to the presence of F508del-CFTR and 20 interactors specific to the presence of wt-CFTR. **(B)** Gene Ontology analysis of the biological processes (BP) enriched in the YAP1 interactome in wt- and F508del-CFTR cells. The biological processes were colored according to manually curated biological categories. Enriched differential BP terms (fold enrichment > 8.5) are represented for YAP1 interactors in wt- and F508del-CFTR cells. Data represented as fold enrichment ($P$ < 0.05). **(C)** Gene set enrichment analysis of the hallmark gene signatures enriched in the YAP1 interactome in wt- and F508del-CFTR cells. Data represented as the $\log_{10}$ of the FDR values ($P$ < 0.05). Two vertical y-axis are shown to account for the different magnitudes of the values and to facilitate data visualization.

F508del-CFTR cells. In addition, we performed a gene set enrichment analysis (GSEA) for hallmark gene sets (Supplemental Data 4 and Fig 5C) as well as oncogenic signature gene sets (Supplemental Data 5 and Fig S5B) to identify specific signaling pathways significantly associated with each dataset.

GO and GSEA analysis yielded consistent and complementary results. GO analysis revealed that the YAP1 interactors present in

F508del-CFTR cells are uniquely enriched in chronic inflammation, ECM and cell-cycle BP terms (Fig 5B), which are major hallmarks of both CF lung disease and fibrotic EMT. Some immune system and apoptosis-related BP terms were also specifically represented in the presence of F508del-CFTR (Fig 5B). In line with this, GSEA analysis showed that the F508del-CFTR–specific YAP1 interactors were enriched in hallmark gene sets like EMT, mitotic spindle (i.e.,

mitosis), and hypoxia as well as gene sets representing EMT-associated signaling through MYC and TGFβ (Fig 5C). Of note, BP terms related to miRNA signaling, barrier function, and differentiation, as well as gene signatures for PI3K/AKT/MTOR signaling, hedgehog signaling and protein secretion were enriched, although modestly, in the YAP1 interactors present in wt-CFTR cells (Fig 5B and C). Other BP terms' categories and hallmark gene signatures which included apical/cell junctions, mTORC1 signaling, G2/M checkpoint, cytoskeleton and stress were not differently represented in the two datasets (Figs 5B and C and S5A). Interestingly, all oncogenic signature gene sets (including a YAP1 conserved signature) were enriched in the YAP1 interactors from F508del-CFTR cells (versus wt-CFTR cells), which also represented a higher number of oncogenes and tumor suppressors (Fig S5B). Importantly, some of the oncogenic signatures, for example, "genes up-regulated in cells over-expressing LEF1 (LEF1_UP.V1_UP)," were exclusive to the YAP1 interactome in the F508del-CFTR cells (Fig S5B and Supplemental Data 5). Overall, F508del-CFTR shifts the YAP1 interactome towards proteins intervening in several fibrotic/EMT–related biological processes in comparison with CFBE cells expressing wt-CFTR cells.

### A multi-omics description of dysregulated YAP1-related EMT pathways in F508del-CFTR cells

The objective of this work was to find new EMT-associated pathways that are specifically related to F508del-CFTR. Thus, as a final step, we assembled a novel PPI network bridging dysfunctional CFTR and EMT through YAP1 and its interactome, elucidated here through the combined functional genomics and proteomics approaches (Fig 1).

To build this PPI network, a subset of proteins of interest was selected based on several criteria (see the Materials and Methods section). Namely, proteins were selected if they were (i) F508del-CFTR-specific YAP1 interactors defining the biological categories and enriched gene sets related to CF and/or EMT, namely, cell cycle, ECM, inflammation (see Supplemental Data 3), MYC signaling, EMT, mitosis, hypoxia, and TGFβ signaling (see Supplemental Data 4); or (ii) F508del-CFTR–specific YAP1 interactors that are oncogenes or define oncogenic signatures (see Supplemental Data 5). Only the proteins in one of these categories (Fig 6A) were considered for further analysis. This subset was still further examined regarding identification of the proteins appearing in more than one category (highlighted with "*" in Fig 6A) and which are also part of the published CFTR interactome (27) (different colors, Fig 6A).

Next, using this list of (EMT/CF-related) proteins that while belonging to the YAP1 interactome, potentially bridge it to dysfunctional CFTR, we used the STRING database (28) to create a predicted PPI network, including CFTR and YAP1 (Fig 6B). Interestingly, most nodes (proteins) showed at least one predicted PPI, suggesting that this is a robust network, supported by extensive public sources (available through the STRING database). Of note, the furthest nodes from YAP1 are only four PPIs away, which supports the fact that indeed we have successfully established a YAP1 interactome. Importantly, and as expected, YAP1 and CFTR are important signaling hubs within the network, showing 4–6 predicted PPIs with other proteins of the network (Fig 6B). However, there are other nodes/proteins that are also major hubs, including

TP53 (tumor protein p53, which shows predicted PPIs with 14 proteins within the network), HSP90AB1 (heat shock protein 90 α family class B member 1, which shows 11), VIM (vimentin, with 10), CAV1 (caveolin-1, showing 8), HSPB1 (heat shock protein β-1, with 8), and PDIA3 (protein disulfide-isomerase A3, also with 8). Several other nodes also show four to six predicted PPIs with other proteins within the network. Interestingly, YAP1 and CFTR are predicted to directly interact, which is mainly due to PubMed text mining findings where both proteins are co-mentioned.

As a last characterization step, we checked the top five hallmark gene sets and oncogenic signature gene sets from the proteins comprising the network. EMT, mitotic spindle (i.e., mitosis), and MYC signaling were again enriched hallmark gene sets (Fig 6C), highlighting the significance of these processes. Regarding the oncogenic signature gene sets, the YAP1 conserved signature was again enriched as well as the LEF1_UP.V1_UP signature (Fig 6C), pointing to a potential role of the Lef1 protein (i.e., a Wnt signaling effector) in F508del-CFTR and YAP1-related EMT.

## Discussion

The goal of this work was to identify potential EMT pathways and protein networks that might be active/dysregulated in response to dysfunctional CFTR and thus affect the overall cellular phenotype including epithelial differentiation and consequently the secretory trafficking machinery. Previously, we have shown that (partial) EMT is active in the presence of mutant CFTR (14), but the identity of EMT-associated factors and pathways involved was still unexplored. We hypothesize that proteins that enhance F508del-CFTR trafficking without affecting wt-CFTR localization would reveal aberrantly regulated EMT pathways. To address this, we followed a multi-omics systems biology approach combining functional genomics and proteomics data, resorting to an siRNA high-throughput screening platform and MS-based interaction proteomics.

To identify EMT-associated pathways activated by the presence of dysfunctional CFTR, we first performed an siRNA screen targeting EMT-related genes using wt- and F508del-CFTR traffic reporter cell lines. This first screen identified several F508del-CFTR-specific traffic enhancer siRNAs, that is, those rescuing F508del-CFTR to the PM without affecting wt-CFTR. The top-hit siRNAs from this primary screen targeted the TFs CEBPB, TWIST1, TEAD4, and YAP1, suggesting that these proteins have an impact on F508del-CFTR traffic. We then validated these top hits using complementary assays and cell models, although the magnitude of F508del-CFTR rescue was relatively low, supporting an indirect effect of these TFs on F508del-CFTR traffic. The exception was C/EBPβ, whose effects were not confirmed probably because of an siRNA off-target effect. Amongst the four top hits, TWIST1 and YAP1 seemed like the most relevant because their expression was significantly increased in the presence of F508del- versus wt-CFTR–expressing cells, most likely being the agents of EMT activation in the presence of dysfunctional CFTR. The importance of these TFs in CF-related EMT was not completely unexpected because we have previously found that mutant CFTR drives partial EMT through a TWIST1-dependent mechanism (14). However, here we found that TWIST1 interacts with both wt- and F508del-CFTR proteins, whereas YAP1 only

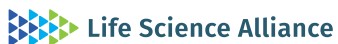

**Figure 6. Data integration points to multiple YAP1-related oncogenic pathways being dysregulated in F508del-CFTR cells.**
**(A)** List of proteins obtained throughout the work that were considered for further analysis and their respective categories. Data were also crossed with a CFTR interactome (27). * Indicates a hit that appears in more than one category. Green indicates overlap with wt-CFTR interactome. Red indicates overlap with F508del-CFTR interactome. Blue indicates overlap with both CFTR interactomes. **(A, B)** Prediction of the protein–protein interaction network for the subset of proteins in (A) was generated using the STRING database (28) (medium level of confidence). The network nodes represent proteins, and the edges represent the predicted functional associations, whereas the thickness of the line indicates the degree of confidence of the interaction. **(C)** Gene set enrichment analysis of the top 5 hallmark gene signatures and oncogenic gene signatures enriched in the top genes/proteins. Data represented as the $\log_{10}$ of the FDR values ($P < 0.05$).

interacts with F508del-CFTR. This suggests that YAP1 plays a more specific/direct role in CF-associated EMT. Importantly, although TWIST1 was found to promote EMT by inhibiting the Hippo pathway (29) and the YAP1/TEAD complex promotes fibrosis by activating the expression of TWIST1 (30), here we did not find an interaction between YAP1 and TWIST1. Thus, based on these findings we set out

to further characterize a potential role for YAP1 in EMT in the context of CF using systems biology. Our data point to TEAD4 being an important YAP1 binding partner in the CF context, which is likely explained by the fact that the YAP1-TEAD association is essential to mediate YAP1-dependent gene transcription, in both homeostasis and disease (23).

YAP1 is a main transcriptional regulator of the mammalian Hippo pathway, an evolutionary conserved signaling pathway (22) which integrates diverse signals (e.g., cell adhesion and polarity, mechanical forces, soluble factors, and various stress signals (31)) to regulate development, homeostasis, and regeneration, but also EMT, fibrosis, and cancer (32). Here, YAP1 emerged as a key TF in CF-related EMT, by showing, not only that it is up-regulated in CF versus non-CF cells, but also that YAP1 specifically interacts with F508del-CFTR impairing its traffic, whereas its KD released F508del-CFTR to the PM and rescued its processing and function.

Importantly, rescue of F508del-CFTR PM traffic by YAP1 KD was additive to that by CFTR highly effective modulator drugs (VX-661+VX-445), indicating independent rescue mechanisms. This not only suggests that YAP1 is a potential novel therapeutic target for CF, but also stresses the well-known need for multiple/combinatory therapeutic strategies targeting the various defects of the mutant channel to achieve full clinical benefit of CF individuals in all aspects, including cancer propensity (33). Consistently, we have previously proposed that successful CFTR modulator drugs need to address not just the rescue of CFTR-mediated ion transport, but also CFTR protective role against EMT to protect individuals with CF from fibrosis and cancer propensity (14).

All these promising results on the relationship between YAP1 and F508del-CFTR motivated us to dissect the pathways bridging this TF and mutant CFTR. To this end a MS-based interaction proteomics approach was followed, to assess differential PPIs established by YAP1 in the presence of F508del- versus wt-CFTR. We found a distinct YAP1 interactome as well as an increased number of YAP1 interacting proteins in F508del- versus wt-CFTR expressing cells. This is in line with data by Pankow et al, who showed that the presence of the F508del mutation disturbs the network of CFTR protein interactions, with mutant CFTR interacting with distinct and numerous other proteins relatively to wt-CFTR (27). It thus appears that, in parallel with remodeling the CFTR interactome, mutant CFTR can also change the landscape of YAP1 interactors. No components of the canonical Hippo pathway were identified here, and the reason for this may be twofold. First, the main roles of YAP1 as a signaling pathway integrator are known to occur independently of the core Hippo pathway (34) and second, the particular MS conditions used here promoted the identification of non-transient versus transitory interactions, being other core Hippo-related proteins among the latter. Importantly, Reilly et al, who found an increased expression of Hippo signaling proteins among the top gene sets enriched in the F508del–CFTR interactome, reported that none were part of the canonical components (35).

Regarding the top biological processes and gene signatures in each dataset, we found that chronic inflammation, ECM-related processes, mitosis, hypoxia, EMT, and MYC and TGFβ signaling were all enriched in the presence of F508del-CFTR, whereas miRNA signaling, barrier function, differentiation, PI3K/AKT/MTOR and hedgehog signaling, and protein secretion were enriched in wt-CFTR cells. This suggests that YAP1 has different roles when wt- or F508del-CFTR are present, in line with its context-specific roles. Moreover, mutant CFTR appears to be enough to change the YAP1 transcriptional program. In particular, most of the processes that are active in F508del-CFTR cells overlap with YAP1 major roles in promoting fibrotic cellular phenotypes, such as myofibroblast

differentiation and increased matrix remodeling potential (32), as well as with its being a central promotor of EMT and tumorigenesis (34). However, what is most striking is that the processes enriched in the F508del-CFTR cells are major hallmarks of CF lung disease. Indeed, CF lungs are characterized by constant chronic inflammation, epithelial remodeling that ultimately leads to fibrosis (2), increased levels of TGF-β1 (36), hypoxia (2), aberrant myofibroblast persistence (37) and excessive collagen I deposition, as well as partially active EMT (14). The fact that the fibrotic/EMT CF pathogenesis phenotypes are in line with the processes enriched in the F508del-CFTR-specific YAP1 interactors is consistent with YAP1 being a central mediator of EMT and fibrosis in CF. In turn, the CF lung environment may further account for the abnormally active YAP1. Because of its oncogenic potential, YAP1 is tightly regulated by the cytoskeleton, ECM and cell–cell contacts (38), with a stiff ECM (39) and lack of cell–cell contacts (40) both resulting in YAP1 nuclear activity. Disruption of epithelial structures, particularly increased ECM and abnormal cell junctions, are present in the CF lung (14, 41, 42) and likely lead to nuclear YAP1 expression which further potentiates fibrosis and EMT.

Last, to address the central question of which YAP1-dependent genes/pathways are involved in CF-related EMT, we integrated our F508del-CFTR-specific EMT/CF-related YAP1 interactors with the CFTR interactome (27). Our results highlighted multiple potentially active EMT-associated pathways in CF, with YAP1 mediating the crosstalk between several of them, but also suggesting co-activation of these signaling pathways. Essentially, five potential EMT pathways/hubs could be involved in mutant CFTR-related EMT (Fig 7): (1) the Hippo pathway (through YAP1/TEAD4), (2) the Wnt pathway (through LEF1), (3) the TGFβ pathway, (4) the p53 pathway, and (5) MYC signaling. Importantly, all these pathways are in the top 10 canonical signaling pathways with frequent genetic alterations that drive carcinogenesis (43). Thus, a simplistic view of a single EMT pathway may not apply to CF-related EMT, which is in line with the extensive network of the F508del-CFTR interactome. Of note, crosstalk occurs among the above pathways. Indeed, MYC is a target of Wnt signaling, while also potentiating LEF1 (44), which in turn can activate the TGFβ signaling pathway (45), and YAP1, Wnt and TGFβ signaling also have several crosstalk points (reviewed in reference 46). In addition, YAP1 also regulates p53 (47) and c-Myc (48). Importantly, mutant CFTR has also been widely related with aberrantly increased TGFβ (36, 49, 50) and Wnt/β-catenin (51, 52) signaling.

Besides these pathways, a novel PPI network was also identified here, although ascertaining the exact influence of the involved proteins in CF still requires investigation. Nevertheless, data in the literature suggest that some of these proteins are important for the pathway's crosstalk and/or have been related to CFTR (Fig 7). Examples include the plasminogen activator inhibitor 1 (PAI-1, SERPINE1), which is a pivotal pro-fibrotic YAP1-TEAD4 transcriptional target (53, 54), is increased in CF lungs and primary airway epithelial cells in conjunction with increased TGFβ signaling (49, 50). PAI-1 also controls the urokinase plasminogen activator (uPA) cancer pathway that is blocked by functional CFTR (13). Importantly, PAI-1 bridges YAP1 (53, 54), Wnt signaling (55), and TGFβ signaling (56), also being induced by p53 (57) and caveolin-1 (58) to promote cellular senescence. Caveolin-1, a component of PM caveolae, is a direct target gene of YAP and TEAD (59) that also regulates YAP1 in

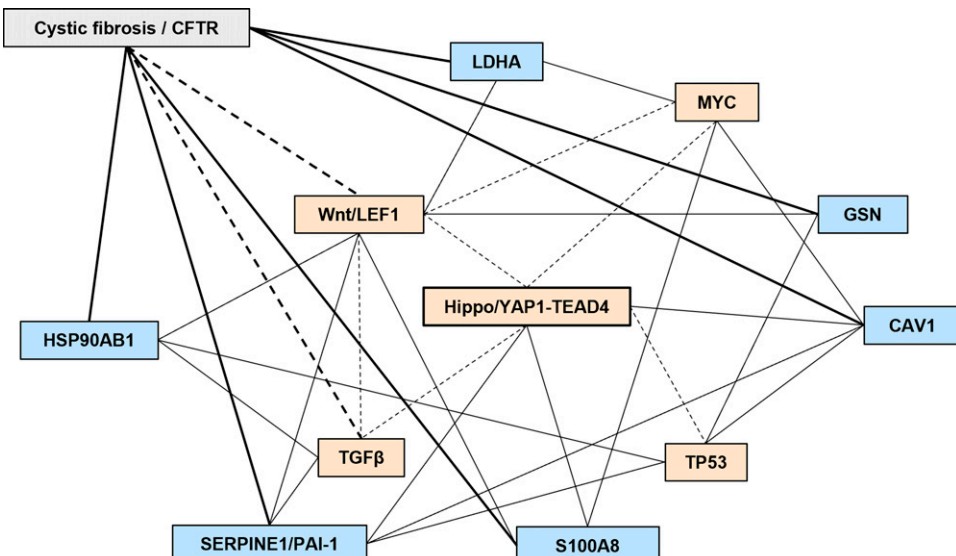

**Figure 7. Data suggest multiple potentially active epithelial–mesenchymal transition-associated pathways in CF, with YAP1 mediating their crosstalk.**
Several pathways and proteins of interested identified by our study are shown as well as literature-based evidence crosstalk between them. Lines indicate evidence of a relationship between a protein and a pathway and dashed lines indicate crosstalk between signaling pathways.

tumorigenesis ([60]). Moreover, caveolin-1 is also associated with c-Myc in prostate cancer ([61]) and KD of CFTR lead to increased inflammation and caveolin-1 levels ([62]).

HSP90AB1 is a molecular chaperone with roles in tumorigenesis ([63]) that activates the Wnt/$\beta$-catenin signaling pathway in gastric cancer ([64]) that also enhances the TGF-$\beta$/SMAD signaling in lung adenocarcinoma (LUAD) ([65]), promoting EMT in both scenarios. In addition, silencing of p53 leads to increased expression of Hsp90 and of its cochaperone activator of heat shock 90 kDa protein ATPase homolog (Aha1) ([66]) whose down-regulation leads to rescue of mutant CFTR to the PM ([67]). Lactate dehydrogenase A (LDHA) was recently found as part of the same oxidative phosphorylation-related gene signature in LUAD together with CFTR, but whereas high levels of CFTR associated with better survival, high LDHA expression was associated with poor prognosis. Interestingly, HSPD1 (another chaperone acting as a network hub) and TWIST1 (one of the links between mutant CFTR and EMT ([14])) were also part of this gene signature and high levels of both proteins also associated with poor survival rates ([68]). LDHA is also an important protein in glycolysis, and this process drives tumor progression through a c-Myc-LDHA axis ([69]) whose levels can be additionally modulated by the Wnt/$\beta$-catenin signaling ([70]). Gelsolin (GSN), an actin severing protein, is described as important for CFTR activation ([71]) but it is also regulated by $\beta$-catenin to promote adenocarcinoma cell migration ([72]). Like HSP90, gelsolin also inhibits p53 signaling ([73]). The neutrophil chemokine S100A8 is an important neutrophil attractor and chronic inflammation mediator in the lungs of CF mice ([74]) which also contributes to colorectal cancer cell survival and migration via the Wnt/$\beta$-catenin pathway and consequent c-Myc expression ([75]). This protein is also modulated by the Hippo pathway in squamous cell carcinoma ([76]).

Finally, there is the mechanistic question of how F508del-CFTR, localized in the ER/ubiquitin-proteasome pathway, interacts with and leads to increased YAP1 translocation into the nucleus. One hypothesis is that YAP1 and F508del-CFTR may interact through an intermediate, likely a chaperone acting at one of the ERQC folding checkpoints (probably the first as it is the main one retaining F508del in the ER ([77])) where mutant CFTR is retained due to its misfolding. The experimental support for this hypothesis is, when we knockdown YAP1, F508del-CFTR is further rescued by VX-445/VX-661 compared with control (scrambled siRNA). This implies that YAP1 (which is up-regulated in CF cells) directly or indirectly inhibits F508del-CFTR from leaving the ER.

Overall, in the present work, we identified several proteins and pathways, most of which directly or indirectly regulated through YAP1, which likely mediate signaling in CF-related EMT. The data revealed several YAP1 and CFTR interactors deserving further investigation on their respective mechanisms of action, which may provide encouraging new therapeutic routes for both CF and CFTR-related tumorigenesis. Encouragingly, because EMT is a highly conserved process across cell types and organ systems, EMT inhibitor drugs originally developed for malignancies like cancer or fibrosis could be repurposed to treatment of CF.

# Materials and Methods

### Cell lines

CFBE cells stably overexpressing wt- and F508del-CFTR ([78]) or overexpressing an inducible mCherry-Flag-CFTR construct (wt- and F508del-CFTR variants) ([20]), as well as Human Bronchial Epithelial Cell Lines (16HBE and 16HBEge) were used throughout this work.

CFBE wt-, F508del-CFTR 16HBE wt-, F508del-, and N1303K-CFTR cells were grown in Minimum Essential Medium Eagle (MEM) with Earl salts and L-glutamine (10-010-CVR; Corning) supplemented with 10% (vol/vol) FBS (10270; Gibco), 1% Pen/Strep (15070; Gibco), and 2.5 $\mu$g/ml puromycin (P8833; Sigma-Aldrich). 16HBE cells were seeded in 24-well plates at the density of 1.2 × 10$^5$ cells/well. On the following day, the medium was changed from 10% to 2% (vol/vol) FBS, and 24 h later, protein was collected for subsequent WB experiment. To achieve polarization, CFBE cells were seeded on

collagen IV pre-coated Transwell permeable supports at a density of 1.25 or 2.5 × 10⁵ cells, depending on the diameter of the filter (6.5 mm or 12 mm insert, Corning 3470 and 3460, respectively). On the following day, the medium was changed from 10% to 2% (vol/vol) FBS to promote differentiation/polarization. The transepithelial electrical resistance (TEER) was measured regularly using a volt-ohmmeter (Millicell-ERS, MER5000001; Millipore).

CFBE mCherry-Flag-CFTR cells were grown in DMEM with 4.5 g/l glucose and L-glutamine (BE12-604F; Lonza) supplemented with 10% (vol/vol) FBS, 10 µg/ml blasticidin (ant-bl-1; InvivoGen), and 2 µg/ml puromycin. 16HBE and 16HBEge cells were provided by CFF.

### CFTR modulator treatments

3 µM VX-445 (S8851; SelleckChem), 5 µM VX-661 (S7059; SelleckChem), or 3 µM VX-809 (S1565; SelleckChem) were added to CFBE cells 48 h after transfection with NCtrl or siYAP1 (1) and left for 24 h. Compounds were dissolved in DMSO which was used as a negative control in the experiment.

### Forward siRNA transfection

CFBE wt- and F508del-CFTR cells were grown to confluence and split (50%). After 24 h, the cells were seeded at an approximate density of 3.2 × 10⁴ cells.cm⁻². Transfection was performed 24 h later by mixing an siRNA solution with Lipofectamine 2000 (11668019; Invitrogen) at a 1:2 ratio in OptiMEM (31985088; Gibco), according to the manufacturer's instructions (i.e., 1 µl of 30 µM siRNA—final concentration of 37.5 nM—with 2 µg of Lipofectamine 2000). Cells were grown in FBS-free media for 24 h before changing the media to MEM supplemented with 10% (vol/vol) FBS. At 72 h post-transfection, protein or RNA were collected.

16HBE-F508del-CFTR cells were seeded at a density of 1.4 × 10⁵ cells.cm⁻² on 24-well plates. Transfection was performed in suspension by mixing a siYAP1 or NCtrl solution with Lipofectamine 3000 (L3000015; Invitrogen) at a 1:0.75 ratio in OptiMEM, according to the manufacturer's instructions (i.e., 1 µl of 30 µM siRNA—final concentration of 37.5 nM—with 0.75 µl of Lipofectamine 3000). 24 h post-transfection, the cells were treated with 3 µM VX-445 (S8851; SelleckChem) and 5 µM VX-661 (S7059; SelleckChem) for 24 h before protein was collected.

### CFTR traffic screen

A high-throughput fluorescence microscopy pipeline for the assessment of the effects of particular genes on CFTR traffic has been previously developed (20). This assay relies on the CFBE mCherry-Flag-CFTR cells, which feature a mCherry tag at the N terminus of CFTR and a Flag tag in its fourth extracellular loop (Fig S6A), facilitating quantification of the total amount of CFTR and amount of CFTR at the PM, respectively. The pipeline allows the quantification (as a modified Z-score) of the impact of an individual gene on CFTR traffic by using siRNAs to KD that specific gene. Treatments are defined as "positive hits" or CFTR traffic enhancers if the siRNA increases PM-CFTR localization by Z > 1 and "negative hits" or CFTR traffic inhibitors if the siRNA decreases PM-CFTR localization by Z < −1. In the present work, a hypothesis-driven siRNA screen was performed with a library containing siRNAs targeting 30 genes known to be associated with EMT and (de)differentiation. The library contained 62 siRNAs: 59 targeted EMT and (de)differentiation genes (2 for most genes) and three were assay controls: a transfection control (siCFTR), a negative control (NCtrl, Negative Control #1, 4390843, Silencer Select; Ambion), and a positive control (siCOPZ1, targeting a subunit protein from COPI vesicles). Importantly, the positive control siCOPZ1 illustrates an F508del-CFTR traffic enhancer (Z = 1.62, Fig S6B and C), whereas treatment with siCFTR greatly decreased the fluorescence signal in almost all cells (Fig S6C), confirming a high-transfection efficiency. The library comprised a total of 83 different treatments, 62 with individual siRNAs, and 21 with two siRNA combinations. The combinations were performed to KD genes which have been reported to act together, for example, YAP1 and TEAD4, which are co-transcriptional activators. The full contents of the library are disclosed in Supplemental Data 1.

### Preparation of siRNA–coated 96-well plates for reverse transfection

96-well plates (655090; Greiner Bio-One) were coated with a custom siRNA library (Silencer Select, Ambion) for solid-phase reverse transfection in a protocol adapted from references 20 and 79. Initially, a transfection mixture of 1,200 µl was prepared by mixing 323 µl of Lipofectamine 2000, 554 µl of a 0.4 M sucrose in OptiMEM solution, and 323 µl of doubly distilled water. Separately, 5 µl of each siRNA at 3 µM added to each well of a conical bottom 96-well plate. Then, 7 µl of the transfection mix were added per well to the siRNA plate. After 20 min incubation at room temperature, 7 µl of 0.2% (wt/vol) gelatin solution were added to the siRNA plate. The total of 19 µl of the final mix (siRNA plus transfection mix plus gelatin) were diluted 1:50 in a 96-well deep well plate using double distilled water. Finally, 50 µl from the diluted mix were applied to each well of the 96-well imaging plates, which were lyophilized and stored in an anhydrous atmosphere. Careful mixing and centrifugation below 30g were performed between steps. This siRNA plate coating was performed with a manual 96-channel pipette (Liquidator96, 17010335; Mettler Toledo) and a VIAFLO electronic 96-channel pipette (6031; Integra).

### CFTR traffic assay

CFBE mCherry-Flag-CFTR cells were grown to confluence and split (50%). After 24 h they were seeded in siRNA-coated 96-well plates (100 µl/well, 5 × 10³ cells) using a Multidrop Combi peristaltic dispenser (5840300; Thermo Fisher Scientific) and antibiotic-free medium. Reverse transfection occurred in the next 72 h. F508del-CFTR expression was induced in the last 48 h and wt-CFTR in the last 24 h of the transfection window, using 1 µg/ml doxycycline (D9891; Sigma-Aldrich).

### Immunostaining

Cells were washed once with cold PBS++ (1× PBS supplemented with 1.1 mM MgCl₂ and 0.7 mM CaCl₂, pH 7.4) after which extracellular Flag-tags were immunodetected (see Table S1) without cell permeabilization for 1 h at 4°C. Cells were then incubated for 20 min at 4°C with 3% (vol/vol) PFA (104003; Merck Millipore), after which they

were transferred to room temperature and incubated for 1 h with anti-mouse Cy5-conjugated secondary antibody (2 mg/ml, A10524; Invitrogen). Finally, the cells were incubated with a Hoechst 33342 solution (1 h, 200 ng/ml, B2261; Sigma-Aldrich) after which they were immersed in PBS++ and incubated overnight before imaging. Between incubations, cells were washed three times with PBS++ using the Tecan Hydrospeed plate washer (INSTHS-02; Tecan). PBS++ was used to prepare all solutions, with antibody solutions additionally containing 1% (wt/vol) BSA (A9647; Sigma-Aldrich). All liquid handling was performed with a Liquidator96 pipette.

### Image acquisition and analysis

Cell imaging was performed at room temperature with an automated widefield Leica DMI 6000B system. A 10x HC PL APO objective (Leica) with a numerical aperture of 0.4 was used. Exposure times at maximum light brightness for Hoechst, mCherry, and Cy5 were 200, 1,300, and 8,000 ms, respectively. The Hoechst channel was used for contrast-based autofocus.

Automated image analysis was performed with open-source software tools (CellProfiler, R) and previously established pipelines for file management (80) and image quantification (20). At least five image fields in triplicate were analyzed for all treatments. Initially, images were background corrected with the dark frame/flat field procedure. Then, quality control (QC) procedures excluded cells which did not significantly express CFTR, had abnormal morphology (e.g., apoptotic cells), or contained a significant number of saturated pixels. Cell-wise fluorescence integration allowed quantifying the CFTR amount in the entire cell (mCherry) or PM (Cy5). Then, an additional QC step excluded plates in which CFTR expression was not decreased by at least 30% in the presence of siCFTR comparing with the NCtrl (i.e., transfection efficiency smaller than 30%). In each well, the median Cy5 fluorescence (PM-CFTR) was calculated and converted into a modified Z-score:

$$Modified\ Z-score\ (Z) = \frac{x - Median_{NCtrl,plate_i}}{sd_{NCtrl,plate_i}},$$

where $x$ is the median of the Cy5 integrated fluorescence of all cells at a given well, and $Median_{NCtrl,plate_i}$ and $sd_{NCtrl,plate_i}$ are the median and SD fluorescence of wells containing the negative control siRNA. The modified Z-score is then averaged across experimental replicates and statistical significance versus the negative control is assessed with a $t$ test. By definition, "positive hits" are the siRNAs increasing PM-CFTR localization by $Z > 1$ and "negative hits" are the siRNAs decreasing PM-CFTR localization by $Z < -1$.

### WB

CFBE or 16HBE cells were washed twice with cold PBS and lysed with sample buffer containing 31.25 mM Tris HCl (30721; Sigma-Aldrich), pH 6.8; 1.5% (vol/vol) sodium dodecyl sulphate (SDS) (15553; Gibco); 10% (vol/vol) glycerol (92025; Sigma-Aldrich); 50 mM dithiothreitol (DTT) (D0632; Sigma-Aldrich), and protease inhibitor cocktail (11697498001; Roche). Benzonase (E1014; Sigma-Aldrich) 25 U/ml was also added to shear the DNA. 25–30 μg of protein were loaded onto polyacrylamide gels (4% for stacking and 7/10% for resolving

gels) to perform SDS/PAGE. Transfer onto polyvinylidene difluoride (PVDF) membranes (IPVH00010; Merck Millipore) was performed using a wet-transfer system. The membranes were blocked for 1 h with 5% (wt/vol) non-fat milk (NFM) in PBS supplemented with Tween 20 (BP337-100; Fisher BioReagents). This was followed by incubation with the primary antibody overnight at 4°C, with gentle shaking. HRP-conjugated secondary antibodies were applied for 1 h at RT. All the antibodies were diluted in the blocking solution. Membrane luminescence was detected on a Chemidoc XRS+ system (170-8265; Bio-Rad). Quantification of band intensity was performed using the Image Lab software (170-9690; Bio-Rad), which integrates peak area. All measurements were normalized against loading controls (calnexin or GAPDH). A list of primary antibodies can be found in Table S1.

### Semi-quantitative PCR (analysis of siRNA KD efficiency)

After siRNA transfection, RNA was extracted using the NZY Total RNA Isolation kit (MB13402; Nzytech) according to manufacturer's instructions. cDNA was generated with NZY M-MuLV Reverse Transcriptase (MB08301; Nzytech). PCR reactions were then performed to evaluate the expression of the respective genes and of housekeeping control GAPDH. Each PCR reaction contained sense and antisense primers (0.2 μM), 20 ng cDNA and NZYTaq II DNA polymerase (MB354; Nzytech). The PCR reactions were carried out according to manufacturer's instructions. cDNA was amplified during 35 cycles for 30 s at 94°C, 30 s at 54–58°C (depending on the primers) and 30 s at 72°C. PCR products were visualized by loading on RedSafe (21141; iNtROM) containing agarose gels. Membrane luminescence was detected on a Chemidoc XRS+ system. Quantification of band intensity was performed using the Image Lab software. All gene expression measurements were normalized against the respective GAPDH expression. Sequences for all primers are available in Table S2.

### qRT-PCR

After CFBE cell polarization, RNA was extracted and cDNA was generated as described above. A mix containing forward and reverse primers, cDNA (5 ng) and 1x Evagreen SsoFast PCR reagent (172-5204; Bio-Rad) was used along with a Bio-Rad CFX96 system. Bio-Rad CFX Maestro 1.0 software was used for analysis. Mean relative transcript levels were calculated by normalizing the gene of interest against the control endogenous gene (GAPDH) and applying the ΔΔCT method where $Fold\ Change\ (FC) = 2^{-\Delta\Delta CT}$. A standard cycle protocol was used for PCR amplification (1 min at 95°C followed by 40 cycles of 10 s at 95°C and 30 s at 60°C). Sequences for the primers were found at Harvard PrimerBank (Table S2). Melt curves confirmed amplification of unique specific products.

### Patch-clamp

Patch-clamp experiments were performed to assess F508del-CFTR function after transfection with siYAP1 (1). CFBE F508del-CFTR cells grown on glass-coated cover slips were transfected with siYAP1 (1) and NCtrl for 72 h. The cover slips were then mounted on the stage of an inverted Zeiss microscope and kept at 37°C. Patch pipettes were filled with a cytosolic-like solution containing: KCl 30 mM, K-gluconate 95 mM, NaH$_2$PO$_4$ 1.2 mM, Na$_2$HPO$_4$ 4.8 mM, EGTA 1 mM,

Ca-gluconate 0.758 mM, MgCl₂ 1.03 mM, D-glucose 5 mM, and ATP 3 mM, pH 7.2. Patch-clamp experiments were performed in the fast whole-cell configuration.

The bath was perfused continuously with Ringer solution: NaCl 145 mM, KH₂PO₄ 0.4 mM, K₂HPO₄ 1.6 mM, D-glucose 5 mM, MgCl₂ 1 mM, and Ca-gluconate 1.3 mM, pH 7.4, containing 50 nM TRAM34 (ab141885; Abcam) at a rate of 8 ml/min. Patch pipettes had an input resistance of 2–4 MΩ and whole cell currents were corrected for serial resistance. Currents were recorded using a patch-clamp amplifier (EPC 7, List Medical Electronics), the LIH1600 interface and PULSE software (HEKA) as well as Chart software (AD Instruments). In regular intervals, membrane voltage (Vc) was clamped in steps of 20 mV from −100 to +100 mV from a holding voltage of −60 mV. Current density was calculated by dividing whole-cell currents by cell capacitance. CFTR was activated with a combination of forskolin (2 $\mu$M) and genistein (25 $\mu$M).

### Immunofluorescence staining of polarized cells (IF)

Polarized CFBE cells were fixed with PFA 4% (vol/vol), permeabilized with Triton X-100 (17-1315-01; Amersham Biosciences) 0.5% (vol/vol), and blocked with BSA 1% (wt/vol) before being removed from their supports using a scalpel. Cells were then incubated overnight at 4°C with primary antibodies, after which a mix of the secondary antibodies and nuclear dye (4 $\mu$g/ml, Methyl Green, 67060; Sigma-Aldrich) was applied for 2 h at RT. Negative controls were performed for each experiment by adding BSA instead of primary antibodies (Fig S7). Filter sections were mounted in a mix of N-propyl gallate (P3130; Sigma-Aldrich) and Glycerol for microscopy (104095; Merck). A list of primary antibodies can be found in Table S1.

Imaging was performed with a Leica TCS SP8 confocal microscope, using HC Plan Apo 20x/0.75 and HC Plan Apo 63x/1.4 objectives. Software used for acquisition was Leica's LAS x, and image processing was performed on ImageJ FIJI (81). FIJI was used to generate average image projections.

### YAP1 fluorescence quantification

Image stacks were exported as TIF for fluorescence quantification. Nuclei were segmented using the Methyl Green channel. A median filter (radius = 7 × 7 × 2 pixels in xyz, or 0.5 × 0.5 × 0.6 $\mu$m) was applied and a manual threshold was applied. Object segmentation was refined with the 3D morphological opening implemented in the MorphoLibJ plugin (https://imagej.net/plugins/morpholibj, radius = 10 × 10 × 0 pixels or 0.7 × 0.7 × 0 $\mu$m). Then, nuclei were declumped using the 3D Watershed Split algorithm from the 3D ImageJ suite (https://imagej.net/plugins/3d-imagej-suite). Seeds were computed by performing ultimate points and 3D dilation (30 × 30 × 0 pixels or 2.1 × 2.1 × 0 $\mu$m) in MorphoLibJ. Manual editing was required to split some objects. The remaining cell cytoplasm was segmented by performing the "Dilate Labels" operation in MorphoLibJ (distance = 250 × 250 × 250 pixels or 17.7 × 17.7 × 74.6 $\mu$m). Cells touching the image border or containing small (<147 $\mu$m³), fragmented, mitotic, or otherwise unrepresentative nuclei were discarded. Finally, YAP1 fluorescence was integrated using the 3D ROI manager and the fraction of fluorescence localizing to the nucleus of each cell was calculated. Each image was summarized by the average nuclear

fraction value of all of its cells. Differences in the average nuclear fraction were tested using *t* test at a significance of 0.05.

### In vitro antibody agarose beads crosslinking

Throughout this work, all immunoprecipitations were preceded by an in vitro antibody agarose bead crosslinking step, to prevent excessively abundant peptides from the antibodies which can generate unwanted interferences. Antibodies were incubated with Pierce Protein G Agarose beads (20398; Thermo Fisher Scientific) at a 1:20 ratio (5 $\mu$l of antibody for 100 $\mu$l of beads) for 1 h rocking. Antibody-bound beads (henceforth referred to simply as beads) were then washed with sodium tetraborate decahydrate (S9640; Sigma-Aldrich), 0.1 M, pH 9, and resuspended again in sodium tetraborate decahydrate with dimethyl pimelimidate (DMP) (21667; Thermo Fisher Scientific) at the final concentration of 20 mM. The DMP and beads were mixed for 30 min at RT, after which the reaction was terminated by washing the beads once with ethanolamine (E0135; Sigma-Aldrich) 0.2 M pH 8 and three times with PBS. Finally, beads were resuspended in PBS with 0.02% (wt/vol) sodium azide (S2002; Sigma-Aldrich) and stored at 4°C until use (no longer than 48 h).

### YAP1, CFTR, and TWIST1 co-immunoprecipitation

CFBE wt- and F508del-CFTR cells were washed three times with cold PBS, incubated in Triton lysis buffer (TBS) (25 mM Tris–HCl pH 7.4, 150 mM NaCl₂, 1% [vol/vol] and Triton X-100, supplemented with protease inhibitors) for 30 min rocking at 4°C, scraped and pelleted. For the immunoprecipitation, the supernatants were collected and pre-cleared for 1 h at 4°C with Protein G agarose beads, followed by incubation with anti-YAP1, anti-CFTR, or anti-TWIST1 crosslinked beads overnight at 4°C. Finally, the beads were washed twice with wash buffer (Tris–HCl 100 mM; NaCl₂ 300 mM) supplemented with 1% (vol/vol) Triton X-100 and twice with wash buffer without Triton. The beads were then resuspended in sample buffer (see above), after which WB was performed as described. Because this was a qualitative experiment, no quantifications were performed. A list of primary antibodies can be found in Table S1.

### YAP1 immunoprecipitation and sample preparation for MS

YAP1 immunoprecipitation was performed as described. Briefly, cells were washed, incubated in TBS, scrapped, and pelleted. Supernatants were then pre-cleared and incubated with anti-YAP1 crosslinked beads, which were washed and resuspended in sample buffer. Approximately 10 $\mu$g of protein were then loaded onto polyacrylamide gels (4% for stacking and 10% for resolving gels) to perform a short gel SDS/PAGE (proteins ran for about 1 cm in the 10% gels). Gels were then washed with ultrapure water to remove the SDS and incubated with PageBlue Protein Staining Solution (24620; Thermo Fisher Scientific) for 1 h with agitation. Gels were washed again with ultrapure water and each stained lane (corresponding to one immunoprecipitation sample) was excised and cut into ~1 mm³ cubes, which were stored in Protein LoBind Tubes (0030108116; Eppendorf) in ultrapure water. Samples were kept at 4°C until nano-liquid chromatography-tandem mass spectrometry (nanoLC-MS/MS) analysis was performed. Gel bands were then excised, reduced in 10 mM DTT (Sigma-Aldrich) for 40 min at 56°C, and alkylated in 55 mM

iodoacetamide (Sigma-Aldrich) for 30 min in the dark. The resulting sample was digested overnight with trypsin (Promega) at 37°C and cleaned up with a C18 column (OMIX C18; Agilent).

### NanoLC–MS/MS analysis and protein identification

NanoLC–MS/MS analysis was performed on an ekspert NanoLC 425 cHiPLC system coupled with a TripleTOF 6600 with a NanoSpray III source (Sciex). Peptides were separated through reversed-phase chromatography (RP-LC) in a trap-and-elute mode. Trapping was performed at 2 $\mu$l/min on a Nano cHiPLC Trap column (Sciex 200 $\mu$m × 0.5 mm, ChromXP C18-CL, 3 $\mu$m, 120 Å) with 100% A for 10 min. The separation was performed at 300 nl/min, on a Nano cHiPLC column (Sciex 75 $\mu$m × 15 cm, ChromXP C18-CL, 3 $\mu$m, 120 Å). The gradient was as follows: 0–1 min, 5% B (0.1% formic acid in acetonitrile; Fisher Chemicals); 1–46 min, 5–35% B; 46–48 min, 35–80% B; 48–54 min, 80% B; 54–57 min, 80–5% B; 57–75 min, and 5% B.

Peptides were sprayed into the MS through an uncoated fused-silica PicoTip emitter (360 $\mu$m O.D., 20 $\mu$m I.D., 10 ± 1.0 $\mu$m tip I.D., New Objective). The source parameters were set as follows: 15 GS1, 0 GS2, 30 CUR, 2.5 keV ISVF, and 100°C IHT. An information dependent acquisition (IDA) method was set with a TOF-MS survey scan of 400–2,000 m/z. The 50 most intense precursors were selected for subsequent fragmentation and the MS/MS were acquired in high sensitivity mode for 40 msec. The obtained spectra were processed and analyzed using ProteinPilot software, with the Paragon search engine (version 5.0; Sciex). A UniProt reviewed database (20,394 entries, accessed on 05/01/2021) containing the sequences of the proteins from Human was used. The following search parameters were set: Iodoacetamide, as Cys alkylation; Trypsin, as digestion; TripleTOF 6600, as the Instrument; gel-based ID as Special factors; Biological modifications as ID focus; Search effort, as thorough; and an FDR analysis. Only the proteins with Unused Protein Score above 1.3 and 95% confidence were considered.

### GO analysis and GSEA

Available bioinformatic tools were used to analyze the lists of YAP1 interactors found in the wt- and F508del-CFTR CFBE cells. Data were submitted to the PANTHER (Protein ANalysis THrough Evolutionary Relationships) Classification System (82) to generate a list of the enriched GO biological process (BP) terms ($P < 0.05$). These were then manually curated and grouped into 11 different categories that included most of the enriched BP terms: apoptosis, barrier function, cell cycle, cell junction, cytoskeleton, differentiation, ECM, immune system, inflammation, miRNA, and stress. Two figures were constructed, one representing all the enriched (fold enrichment > 15) BP terms found for the YAP1 interactors on both cell lines (Fig S4B), and another showing the enriched (fold enrichment > 8.5) differential BP terms found for each cell line (Fig 4B).

Data were also submitted to GSEA (83), particularly to the Molecular Signatures Database (MSigDB) to identify gene sets differentially expressed by the YAP1 interactors in the presence of wt- or F508del-CFTR ($P < 0.05$). Data were analyzed regarding hallmark gene sets (MSigDB gene sets that represent well-defined biological states or processes) as well as oncogenic signature gene sets (generated from microarray gene expression data from cancer gene perturbations).

### PPI networks

To assemble a novel PPI network bridging (dysfunctional) CFTR and EMT through YAP1, a subset of genes/proteins of interest derived from the MS analyses, was established. This subset consisted of genes/proteins that met at least one of the following criteria, namely, if they were (ii) F508del-CFTR-specific YAP1 interactors defining unique GO BP terms related to CF and/or EMT, that is, from the manually curated biological categories ECM and inflammation (see Supplemental Data 3); (ii) F508del-CFTR–specific YAP1 interactors defining gene sets enriched in F508del-CFTR cells and related to CF and/or EMT, that is, MYC signaling, EMT, mitotic spindle (mitosis), hypoxia, and TGF$\beta$ signaling (see Supplemental Data 4); and (iii) F508del-CFTR-specific YAP1 interactors that were unique oncogenes or defined oncogenic signatures in F508del-CFTR cells (see Supplemental Data 5). This subset comprised 65 proteins (Fig 6A), excluding YAP1 and CFTR. To investigate which, out of these interactors, were the more relevant in the context of CF, they were filtered by one of two criteria: (i) proteins that appeared in more than one category (represented by "*" in Fig 6A) and/or (ii) proteins that are part of the published CFTR interactome (27) (represented by the different colors in Fig 6A). This list of proteins of interest from the YAP1 interactome comprised 30 high confidence proteins (Fig 6A), for which predictions of PPI networks were generated using the STRING database (28) (medium level of confidence). STRING integrates all publicly available sources of PPI information and complements these with computational predictions.

### Antibodies, primers, and siRNAs

Information regarding the siRNAs used in the present study are available in Supplemental Data 1. A list of primary antibodies used in both IF and WB can be found in Table S1. Sequences for the primers used are listed in Table S2.

### Statistical analyses

Data are presented as mean ± SEM, mean ± SD, and median ± SEM, as indicated in the figure legends. Graphical representation in the form of data points was not pursued since most of the data was normalized to the control condition (Fig S8). Number of replicates is also present in all figure legends and indicates how many times the experiment was performed (i.e., n = 3 indicates experiments performed in triplicate). $t$ test for unpaired samples was performed for statistical analyses. Either Prism 6 software (GraphPad, Inc.) or open-source software R were used for graph design and statistical analyses. R-based statistical analysis is described in the "Image acquisition and analysis" sections. R-based graph design was performed with the ggplot2 (84) and VennDiagram (85) packages. Significant differences were defined for $P < 0.05$.

## Data Availability

All the datasets from the present work can be found as Supplementary materials (Supplemental Data S1–S5).

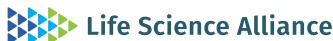 Life Science Alliance

# Supplementary Information

# Acknowledgements

This work was supported by UID/MULTI/04046/2020 and UIDP/04046/2020 center grants from FCT/MCTES, Portugal (to BioISI), and research grants (to MD Amaral): "DiffTarget" (FCT/POCTI/PTDC/BIM-MEC/2131/2014) and SRC 013 from CF Trust-UK. MC Quaresma was a recipient of PD/BD/114389/2016 fellowship from BioSys PhD program PD/00065/2012 from FCT (Portugal). The CFTR traffic screen was performed at the Faculty of Sciences of the University of Lisboa Microscopy Facility, a node of the Portuguese Platform of BioImaging, reference PPBI-POCI-01-0145-FEDER-022122. Mass spectrometry data were generated by the Mass Spectrometry Unit (UniMS), ITQB/iBET, Oeiras, Portugal.

## Author Contributions

MC Quaresma: conceptualization, data curation, formal analysis, investigation, methodology, and writing—original draft, review, and editing.
HM Botelho: conceptualization, data curation, formal analysis, methodology, and writing—review and editing.
I Pankonien: conceptualization, data curation, formal analysis, investigation, methodology, and writing—review and editing.
CS Rodrigues: formal analysis and investigation.
MC Pinto: formal analysis, investigation, and methodology.
PR Costa: formal analysis, investigation, and methodology.
A Duarte: formal analysis and methodology.
MD Amaral: conceptualization, funding acquisition, and writing—review and editing.

## Conflict of Interest Statement

The authors declare that they have no conflict of interest.

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
