## [Reviewer comments · Life Science Alliance]

Life Science Alliance

Exploring YAP1-centred Networks Linking Dysfunctional CFTR to Epithelial-Mesenchymal Transition

Margarida Quaresma, Hugo Botelho, Ines Pankonien, Cláudia Rodrigues, Madalena Pinto, Pau Costa, Aires Duarte, and Margarida Amaral

DOI: <https://doi.org/10.26508/lsa.202101326>

Corresponding author(s): Margarida Amaral, University of Lisbon

Review Timeline:

Submission Date:	2021-11-30
Editorial Decision:	2022-01-18
Revision Received:	2022-03-24
Editorial Decision:	2022-04-12
Revision Received:	2022-04-20
Accepted:	2022-04-20

Transaction Report:

January 18, 2022

Re: Life Science Alliance manuscript #LSA-2021-01326-T

Prof Margarida D Amaral
University of Lisboa
Faculty of Sciences, BioISI - Biosystems & Integrative Sciences Institute
Campo Grande, C8 bdg
Lisboa, Estremadura 1749-016
Portugal

Dear Dr. Amaral,

Thank you for submitting your manuscript entitled "YAP1 is a Key Player Linking Dysfunctional CFTR to Epithelial-Mesenchymal Transition in Cystic Fibrosis" to Life Science Alliance. The manuscript was assessed by expert reviewers, whose comments are appended to this letter. We invite you to submit a revised manuscript addressing the Reviewer comments.

Thank you for this interesting contribution to Life Science Alliance. We are looking forward to receiving your revised manuscript.

Sincerely,

B. MANUSCRIPT ORGANIZATION AND FORMATTING:

Reviewer #1 (Comments to the Authors (Required)):

The reviewed manuscript employs a systems biology approach to address the non-canonical effects of CFTR mutations on epithelial biology. Namely, the authors used functional genomics and interactive proteomics to explore how the $\Delta F508$ CFTR mutation contributes to epithelial-mesenchymal transition (EMT). The authors hypothesize that proteins that enhanced $\Delta F508$ trafficking without changing WT-CFTR localization would reveal aberrantly regulated EMT pathways. To this end, the authors employed an siRNA screen of EMT-associated genes and evaluated $\Delta F508$ localization. Four candidates (YAP1, TEAD4, TWIST1, and CEBPB) were then validated as potential correctors of CFTR localization. The authors subsequently performed proteomic analysis of YAP1 interactions comparing cells with wildtype and $\Delta F508$. Lastly, the authors performed pathway analyses to suggest a model by which YAP1 and $\Delta F508$ could contribute to enhanced EMT in CF. Experiments performed were sound. However, sometimes interpretations over-reached the data.

Major comments.

1. The title is too conclusive for the data. Specifically, the second half of the title would suggest that the manuscript directly studied EMT. The title should reflect that YAP1 links CFTR to EMT bioinformatic networks.
2. The authors state that $\Delta F508$ CFTR increases YAP1 translocation into the nucleus. On the other hand, YAP1 interacts with $\Delta F508$ CFTR, which is known to be localized in the ER/ubiquitin-proteasome pathway. It is worth expanding on how the authors envision YAP1's trafficking in the discussion.
3. Nuclear localization experiments (Fig 4F-G) may be subject to artifacts from staining/thresholding and are more correlative than quantitative. Lack of follow-up experiments or controls warrants softened conclusions. Alternatively, a western blot using nuclear fractions would be more rigorous.
4. The authors find that CFTR dysfunction affects YAP-1, and knocking down YAP-1 restores CFTR function. Did the authors find that modulator treatment altered YAP-1 expression (Blots in 4D)? Any finding from this is worth reporting and adding to the discussion/model.

Minor comments.

1. Statistics are hard to appreciate. For transparency and reproducibility, please be more specific. For example, $n = 3$ passages of cells or $n = 3$ experiments performed in triplicate.
2. There are some occurrences of the term "constitutively active" for YAP activity. This phrase is often used for gain-of-function mutations where activation cannot be reversed. Please substitute a new term for constitutively active.
3. The IP experiment in 4A doesn't have a particularly strong band for YAP1/CFTR. It may be worth mentioning that this finding is later confirmed by mass-spec.
4. Show data points in place of bar graphs. n is low and therefore data points will not obscure the graph.

Reviewer #2 (Comments to the Authors (Required)):

The article is entitled « YAP1 is a key player linking dysfunctional CFTR to epithelial-mesenchymal transition in Cystic Fibrosis » and submitted by Quaresma et al.

Based on studies reporting the correlation between CF and an increased risk of cancer, as well as the description of CFTR as a tumor suppressor, the authors proposed to identify potential EMT pathways and protein network that might be active or dysregulated in response to dysfunctional CFTR

Using an elegant airway epithelial cell model consisting of the CFBE cells expressing an inducible mCherry-Flag-CFTR (wt- or F508del-CFTR), they used high-throughput fluorescence microscopy screening pipeline to quantify CFTR traffic in response to inactivation of EMT genes using a library of siRNAs. They identified 4 hits whose knockdown selectively rescued F508del-CFTR traffic among them YAP1 which they described as only interacting with F508del-CFTR and not with wt-CFTR. In addition to rescue a plasma membrane expression of F508del-CFTR, the authors showed that YAP1 knockdown had an additive effect to the vertex compound to increase this membrane localization of F508del-CFTR. Finally, by mass spectrometry-based interaction proteomics and bioinformatics, the authors described the YAP1 interactome in wt- and F508del-CFTR expressing cells, and proposed some dysregulated YAP1-related EMT pathways in F508del-CFTR cells.

Although this study, whose goal is to elucidate how CFTR can affect a process such as EMT, is conducted elegantly, the referee has several concerns with some points.

The first one is that the role of YAP1 needs to be validated in another CF cell model in order to verify that the results are not cell line-dependent, and in airway cells with a mutation other than F508del to be able to propose that YAP is a key player in the epithelial-mesenchymal transition in CF.

It is not clear why the authors restricted the list of hits to 4, and why they didn't interest to genes inhibiting the wt-CFTR traffic such as LSM14B, SNAI1 or TAZ.

Why do the authors consider that YAP and TEAD4 inhibition do not affect wt-CFTR traffic when Z scores are less than -1? (siYAP1: -1.2; siTEAD4: -1.43; SiTEAD4+siYAP1: -3.01) ?

Figure 2E/F: What about TWIST in wt-CFTR and F508del-CFTR cells? it is not shown in the figure although the authors mention it in the text.

Figure 4A: from the blot, there is no evidence for an interaction between YAP1 and CFTR, neither wild type nor F508del. What about potential interaction between TEAD4, CEBPB and CFTR?

Figure 4C: we can't see the Band C in VX-661 and VX-809 conditions. It is important to analyze the functionality of the F508del-CFTR channel (Ussing, Patch Clamp,...) rather than the expression of band C.

Figure 4F/G: it is difficult to conclude to an accumulation of YAP1 in the nuclei of F508del-CFTR cells from immunofluorescent pictures. The authors should study the quantitative expression of YAP1 by western blotting on total cellular extracts, and cytosolic and nuclear fractions.

Dataset s2: it is surprising to not find CFTR in the list of the YAP1 interactors in F508del-CFTR cells.

Reviewer #1

The reviewed manuscript employs a systems biology approach to address the non-canonical effects of CFTR mutations on epithelial biology. Namely, the authors used functional genomics and interactive proteomics to explore how the $\Delta F508$ CFTR mutation contributes to epithelial-mesenchymal transition (EMT). The authors hypothesize that proteins that enhanced $\Delta F508$ trafficking without changing WT-CFTR localization would reveal aberrantly regulated EMT pathways. To this end, the authors employed an siRNA screen of EMT-associated genes and evaluated $\Delta F508$ localization. Four candidates (YAP1, TEAD4, TWIST1, and CEBPB) were then validated as potential correctors of CFTR localization. The authors subsequently performed proteomic analysis of YAP1 interactions comparing cells with wildtype and $\Delta F508$. Lastly, the authors performed pathway analyses to suggest a model by which YAP1 and $\Delta F508$ could contribute to enhanced EMT in CF. Experiments performed were sound. However, sometimes interpretations over-reached the data.

Our response:

We are grateful to the reviewer for the positive appreciation of our work and for the helpful comments.

Major comments.

1. *The title is too conclusive for the data. Specifically, the second half of the title would suggest that the manuscript directly studied EMT. The title should reflect that YAP1 links CFTR to EMT bioinformatic networks.*

Our response:

We have changed the title to better convey the contents of our study.

2. *The authors state that $\Delta F508$ CFTR increases YAP1 translocation into the nucleus. On the other hand, YAP1 interacts with $\Delta F508$ CFTR, which is known to be localized in the ER/ubiquitin-proteasome pathway. It is worth expanding on how the authors envision YAP1's trafficking in the discussion.*

Our response:

One hypothesis that YAP1 and F508del-CFTR may interact through an intermediate, likely a chaperone acting at one of the ERQC folding checkpoints (probably the 1st as it is the main one retaining F508del in the ER [PMID: 15923638]) where mutant CFTR is retained due to its misfolding. The experimental support for this hypothesis is, when we knockdown YAP1, F508del-CFTR is further rescued by VX445/VX-661 compared to control (scrambled siRNA). This implies that YAP1 (which is upregulated in CF cells) directly or indirectly inhibits F508del-CFTR from leaving the ER. We have now added a paragraph on this matter the discussion (page 14, paragraph 3).

3. *Nuclear localization experiments (Fig 4F-G) may be subject to artifacts from staining/thresholding and are more correlative than quantitative. Lack of follow-up experiments or controls warrants softened conclusions. Alternatively, a western blot using nuclear fractions would be more rigorous.*

Our response:

We agree that a Western blot using nuclear fractions would be a good complementary technique. However, YAP1, being a transcription factor, has a well-documented nuclear localization. We used a standard (and optimized) immunofluorescence and image acquisition protocol, and we performed the experiments in different biological replicates. Additionally, ZO-1 stainings in Fig.4F attest to the quality of the stainings. Moreover, our quantification method only applies a threshold to isolate and identify

the nuclei, and it does so on the nuclear staining channel, thus any thresholding artifacts are unlikely. The analysis evaluating the YAP1 expression in all the nuclei of an image, is also very robust, as it results in a summarized average nuclear fraction value of all its cells. These experimental details are described in the Methods section (“YAP1 fluorescence quantification”). This analysis was also applied to several imaging fields across different biological replicates. Nonetheless, to support our claims, we have prepared a new figure including the negative controls for inspection by the reviewers (see “Data for inspection”).

4. The authors find that CFTR dysfunction affects YAP-1, and knocking down YAP-1 restores CFTR function. Did the authors find that modulator treatment altered YAP-1 expression (Blots in 4D)? Any finding from this is worth reporting and adding to the discussion/model.

Our response:

We have now quantified the YAP1 expression in the presence of CFTR modulators, but although there was a slight decrease in YAP-1 upregulation, changes were not significant. However, we have now added these quantifications to the Fig.S3.

Minor comments.

1. Statistics are hard to appreciate. For transparency and reproducibility, please be more specific. For example, n = 3 passages of cells or n = 3 experiments performed in triplicate.

Our response:

We have clarified this information in the Methods section (“Statistical analyses” section).

2. There are some occurrences of the term "constitutively active" for YAP activity. This phrase is often used for gain-of-function mutations where activation cannot be reversed. Please substitute a new term for constitutively active.

Our response:

We have now replaced this term with “abnormally activated”.

3. The IP experiment in 4A doesn't have a particularly strong band for YAP1/CFTR. It may be worth mentioning that this finding is later confirmed by mass-spec.

Our response:

In fact, the YAP1/CFTR interaction bands are faint, suggesting a weak interaction, and this interaction could not be confirmed by MS. This is likely due to the fact that “the particular MS conditions used here promoted the identification of non-transient vs transitory interactions” (as mentioned in the Discussion, page 12, paragraph 3). However, the results from the IP were very consistent and highly reproducible.

4. Show data points in place of bar graphs. n is low and therefore data points will not obscure the graph.

Our response:

We have included a few examples of the manuscript’s graphs with data points in the “Data for inspection” file. However, particularly since most of the data is normalized to the negative controls, we do not consider this representation to add value to the existing one. The only exception is Fig. 4G where there is a greater distribution of the data points and thus these are shown on the graph.

Reviewer #2

The article is entitled « YAP1 is a key player linking dysfunctional CFTR to epithelial-mesenchymal transition in Cystic Fibrosis" and submitted by Quaresma et al.

Based on studies reporting the correlation between CF and an increased risk of cancer, as well as the description of CFTR as a tumor suppressor, the authors proposed to identify potential EMT pathways and protein network that might be active or dysregulated in response to dysfunctional CFTR

Using an elegant airway epithelial cell model consisting of the CFBE cells expressing an inducible mCherry-Flag-CFTR (wt- or F508del-CFTR), they used high-throughput fluorescence microscopy screening pipeline to quantify CFTR traffic in response to inactivation of EMT genes using a library of siRNAs. They identified 4 hits whose knockdown selectively rescued F508del-CFTR traffic among them YAP1 which they described as only interacting with F508del-CFTR and not with wt-CFTR. In addition to rescue a plasma membrane expression of F508del-CFTR, the authors showed that YAP1 knockdown had an additive effect to the vertex compound to increase this membrane localization of F508del-CFTR. Finally, by mass spectrometry-based interaction proteomics and bioinformatics, the authors described the YAP1 interactome in wt- and F508del-CFTR expressing cells, and proposed some dysregulated YAP1-related EMT pathways in F508del-CFTR cells.

Although this study, whose goal is to elucidate how CFTR can affect a process such as EMT, is conducted elegantly, the referee has several concerns with some points.

Our response:

We are grateful to the reviewer for the positive appreciation of our work and for the helpful comments.

1. The first one is that the role of YAP1 needs to be validated in another CF cell model in order to verify that the results are not cell line-dependent, and in airway cells with a mutation other than F508del to be able to propose that YAP is a key player in the epithelial-mesenchymal transition in CF.

Our response:

We have now checked the levels of YAP1 in the bronchial epithelial cell lines 16HBE and 16HBEge expressing wt-, F508del- and N1303K-CFTR. We show that YAP1 expression is also significantly increased upon CFTR dysfunction in this second airway cell model, regardless of CFTR mutation. We have also confirmed the additive effect of YAP1 KD and CFTR modulator treatment in an additional model. These data have now been added as Fig.S3 and referred to in the Results section (page 7, paragraphs 1,2).

2. It is not clear why the authors restricted the list of hits to 4, and why they didn't interest to genes inhibiting the wt-CFTR traffic such as LSM14B, SNAI1 or TAZ.

Our response:

We explain our rationale in the beginning of the first Results section (page 5, paragraph 2), stating that "We hypothesized that EMT-related genes whose genetic inactivation had no effect on wt-CFTR traffic but enhanced/rescued F508del-CFTR traffic would likely integrate EMT pathways aberrantly active in response to dysfunctional CFTR". Thus, the four top hits were selected based on the rational of choosing high confidence hits targeting EMT-related gene products whose KD selectively rescues F508del-CFTR traffic. We agree with the reviewer that other findings from this work would be interesting to explore, but it would be impossible to study them all in a single study. However, although these genes were beyond the scope of the present work, we do disclose our entire data sets so that other studies can follow-up on our other findings.

3. *Why do the authors consider that YAP and TEAD4 inhibition do not affect wt-CFTR traffic when Z scores are less than -1? (siYAP1: -1.2; siTEAD4: -1.43; SiTEAD4+siYAP1: -3.01) ?*

Our response:

Due to the already considerable extension of the manuscript, we did not go into much detail about this, but indeed, two different combinations targeting siYAP1+siTEAD4 and siTEAD4+siTAZ (another major Hippo player), emerge as high confidence wt-CFTR traffic inhibitors (see Dataset S1 for details). We have added a sentence to highlight this (page 6, paragraph 1). However, we consider that these results also support of the main message that we are conveying, i.e., that the Hippo signalling (particularly YAP1 and TEAD4) is specifically dysregulated in F508del- vs wt-CFTR expressing cells. Thus, we have not elaborated further on this matter.

4. *Figure 2E/F: What about TWIST in wt-CFTR and F508del-CFTR cells? it is not shown in the figure although the authors mention it in the text.*

Our response:

As we mention in the text: “We have also previously found TWIST1 to be significantly increased in F508del-CFTR expressing cells [PMID: 33106471]” (page 6, paragraph 2). These results have already been published and this is why that are not in Fig. 2E,F.

5. *Figure 4A: from the blot, there is no evidence for an interaction between YAP1 and CFTR, neither wild type nor F508del. What about potential interaction between TEAD4, CEBPB and CFTR?*

Our response:

Albeit faint, the bands demonstrating a YAP1-F508del-CFTR are present (signalled by arrowheads). These results were very consistent and highly reproducible. Regarding TEAD4 and CEBPB, as explained in the manuscript: “Among the 4 top hits of the traffic screen, only protein expression levels of transcription factors (TFs) YAP1 and TWIST1 were significantly increased in F508del- vs wt-CFTR expressing cells, suggesting that any aberrant EMT triggered by F508del-CFTR might be mediated by one or both of these TFs.” (page 7, paragraph 1). Thus, since they did not meet this criterion, the interactions of TEAD4 and CEBPB with CFTR were not further evaluated.

6. *Figure 4C: we can't see the Band C in VX-661 and VX-809 conditions. It is important to analyze the functionality of the F508del-CFTR channel (Ussing, Patch Clamp,...) rather than the expression of band C.*

Our response:

Albeit not as strong as the band C in response to VX-445+VX-661, both the VX-661 and VX-809 treatments clearly result in the presence of band C – it is present around 180kDa and signalled by the arrowhead stating “band C”. In fact, in Fig. 4D (not Fig. 4C), the only treatment for which band C is absent is the treatment with DMSO in cells transfected with NCtrl. We agree that the functionality of the F508del-CFTR channel is important, and this is why we evaluated the impact of YAP1 knockdown on its function by patch-clamp (Fig.4B,C). Since we had previously shown that the presence of band C (detected by Western Blot) was in agreement with the functional data we did not consider it necessary to repeat the experiment with the CFTR modulator drugs.

7. *Figure 4F/G: it is difficult to conclude to an accumulation of YAP1 in the nuclei of F508del-CFTR cells from immunofluorescent pictures. The authors should study the quantitative expression of YAP1 by western blotting on total cellular extracts, and cytosolic and nuclear fractions.*

Our response:

Although we agree that a Western blot using cellular fractions would be a good complementary technique, we consider that our YAP1 staining is specific and that our quantification method is very sensitive. Our quantification technique isolates and identifies the nuclei on the nuclear staining channel, and then evaluates the YAP1 expression in all the nuclei of an image, resulting in a summarized average nuclear fraction value of all of its cells. We used different biological replicates and several imaging fields in order to produce robust data. These experimental details are described in the Methods section (“YAP1 fluorescence quantification”). Nonetheless, to support our claims on the quality of the YAP1 stainings, we have prepared a new figure including the negative controls for inspection by the reviewers (see “Data for inspection”).

8. Dataset s2: it is surprising to not find CFTR in the list of the YAP1 interactors in F508del-CFTR cells.

Our response:

We do mention in the manuscript that the YAP1/CFTR interaction bands are faint, thus likely suggesting a weak interaction. This, plus the fact that “the particular MS conditions used here promoted the identification of non-transient vs transitory interactions” (as mentioned in the Discussion, page 12, paragraph 3) likely justify why we did not find CFTR in the list of YAP1 interactors in F508del-CFTR cells.

April 12, 2022

RE: Life Science Alliance Manuscript #LSA-2021-01326-TR

Dr. Margarida D Amaral
University of Lisbon
Faculty of Sciences, BioISI - Biosystems & Integrative Sciences Institute
Campo Grande, C8 bdg
Lisboa, Estremadura 1749-016
Portugal

Dear Dr. Amaral,

Thank you for submitting your revised manuscript entitled "Exploring YAP1-centred Networks Linking Dysfunctional CFTR to Epithelial-Mesenchymal Transition". We would be happy to publish your paper in Life Science Alliance pending final revisions necessary to meet our formatting guidelines.

- please add an alternate abstract/summary blurb to the main manuscript and our system
- please add a separate section with your figure legends directly after the references; this section should include figure legends for your main figures, supplementary figures, and tables
- please upload your main and supplementary figures as single files, labeled appropriately
- please use the [10 author names, et al.] format in your references (i.e. limit the author names to the first 10)
- please add the Twitter handle of your host institute/organization as well as your own or/and one of the authors in our system
- please make sure the author order in the manuscript and the system match
- please make sure all authors contributions are listed in the author contributions section
- please check your figure 5 legend and label the panels correctly
- The figures in the "Data for Inspection" file should be made into Supplementary figures.
- please add a Data Availability section to indicate any publicly deposited datasets
- The Materials and Methods in the Supplemental Information should be incorporated into the main Materials and Methods section of the paper. There is no limit for this section.
- please add a scale bar for the left panels in Figure S6C

A. FINAL FILES:

B. MANUSCRIPT ORGANIZATION AND FORMATTING:

Sincerely,

Reviewer #1 (Comments to the Authors (Required)):

The authors addressed most concerns, and current the contents are appropriate for publication in this journal.

The reviewer previously suggested that the data points be shown for the general audience, not for the reviewer, but leave it to the authors' and editor's discretion if they choose not to show them.

The authors did not address both reviewers' concerns about the nuclear localization experiment. The reviewer is ambivalent about how the authors addressed the concern and caution that the opinion that this finding could have been more rigorously tested may not be limited to the two reviewers.

April 20, 2022

RE: Life Science Alliance Manuscript #LSA-2021-01326-TRR

Dr. Margarida D. Amaral
University of Lisbon
Faculty of Sciences, BiolSI - Biosystems & Integrative Sciences Institute
Campo Grande, C8 bdg
Lisboa, Estremadura 1749-016
Portugal

Dear Dr. Amaral,

Thank you for submitting your Research Article entitled "Exploring YAP1-centred Networks Linking Dysfunctional CFTR to Epithelial-Mesenchymal Transition". It is a pleasure to let you know that your manuscript is now accepted for publication in Life Science Alliance. Congratulations on this interesting work.

DISTRIBUTION OF MATERIALS:

Again, congratulations on a very nice paper. I hope you found the review process to be constructive and are pleased with how the manuscript was handled editorially. We look forward to future exciting submissions from your lab.

Sincerely,
